# RPG acts as a central determinant for infectosome formation and cellular polarization during intracellular rhizobial infections

**Beatrice Lace[1]\*, Chao Su[1], Daniel Invernot Perez[1], Marta Rodriguez-Franco[1], Tatiana Vernié[2], Morgane Batzenschlager[1], Sabrina Egli[1], Cheng-Wu Liu[3], Thomas Ott[1,4]\***

[1]University of Freiburg, Faculty of Biology, Freiburg, Germany; [2]LRSV, Université de Toulouse, CNRS, UPS, INP Toulouse, Castanet-Tolosan, France; [3]School of Life Sciences, Division of Life Sciences and Medicine, MOE Key Laboratory for Membraneless Organelles and Cellular Dynamics, University of Science and Technology of China, Hefei, China; [4]CIBSS – Centre of Integrative Biological Signalling Studies, University of Freiburg, Freiburg, Germany

**\*For correspondence:**
beatrice.lace@biologie.uni-freiburg.de (BL);
Thomas.Ott@biologie.uni-freiburg.de (TO)

**Competing interest:** The authors declare that no competing interests exist.

**Abstract** Host-controlled intracellular accommodation of nitrogen-fixing bacteria is essential for the establishment of a functional Root Nodule Symbiosis (RNS). In many host plants, this occurs via transcellular tubular structures (infection threads - ITs) that extend across cell layers via polar tip-growth. Comparative phylogenomic studies have identified *RPG* (*RHIZOBIUM-DIRECTED POLAR GROWTH*) among the critical genetic determinants for bacterial infection. In *Medicago truncatula*, *RPG* is required for effective IT progression within root hairs but the cellular and molecular function of the encoded protein remains elusive. Here, we show that RPG resides in the protein complex formed by the core endosymbiotic components VAPYRIN (VPY) and LUMPY INFECTION (LIN) required for IT polar growth, co-localizes with both VPY and LIN in IT tip- and perinuclear-associated puncta of *M. truncatula* root hairs undergoing infection and is necessary for VPY recruitment into these structures. Fluorescence Lifetime Imaging Microscopy (FLIM) of phosphoinositide species during bacterial infection revealed that functional RPG is required to sustain strong membrane polarization at the advancing tip of the IT. In addition, loss of RPG functionality alters the cytoskeleton-mediated connectivity between the IT tip and the nucleus and affects the polar secretion of the cell wall modifying enzyme NODULE PECTATE LYASE (NPL). Our results integrate RPG into a core host machinery required to support symbiont accommodation, suggesting that its occurrence in plant host genomes is essential to co-opt a multimeric protein module committed to endosymbiosis to sustain IT-mediated bacterial infection.

## Editor's evaluation

This work addresses a fundamental question in symbiosis, placing a classic nodulation defective mutant (rpg) into a plausible protein complex and establishing a hierarchy of "infectosome" assembly. It will be of particular interest to cell biologists and those studying host-microbe interactions. The study includes compelling microscopy data for subcellular localization of components during the establishment and maintenance of infection and includes new FLIM-based imaging techniques to distinguish signals from closely associated domains in plant cells.

## Introduction

Legumes have evolved the capacity to maintain a mutualistic association with endosymbiotic nitrogen-fixing rhizobia, hosting them in specialized lateral root organs, the nodules, where bacteria convert atmospheric nitrogen ($N_2$) to ammonia ($NH_4^+$) and release it to the plant in exchange for carbohydrates. By granting them access to an additional source of nitrogen, the RNS allows plants to overcome nitrogen limitations occurring in terrestrial ecosystems and agriculturally used sites. Thus, the transfer of this ability to cereals has been identified as one possible strategy to reduce the use of environmentally harmful and expensive industrial fertilizers (*Bloch et al., 2020*; *Mus et al., 2016*; *Oldroyd and Dixon, 2014*; *Pankievicz et al., 2019*; *Rogers and Oldroyd, 2014*). Such an engineering approach, however, requires an in-depth understanding of the molecular machinery allowing rhizobial infections of legume roots.

In most cases, this host-controlled process requires a molecular dialogue between the partners, which involves the perception of bacterial lipo-chitooligosaccharide (LCO) molecules called Nod factors (NFs) by plasma membrane-resident receptor complexes including the *M. truncatula* LysM receptor-like kinase NFP and its *Lotus japonicus* ortholog NFR5 (*Limpens et al., 2003*; *Radutoiu et al., 2003*). Symbiont recognition triggers the onset of a signaling cascade involving the generation and decoding of nuclear-associated calcium oscillations, which mediates symbiosis-related gene expression (*Singh et al., 2014*). Interestingly, the same pathway is activated by the perception of diffusible signals produced by arbuscular mycorrhizal (AM) fungi, including both LCOs and short-chain chitooligosaccharides (COs), although these are perceived by a different combination of receptors (*Feng et al., 2019*; *Sun et al., 2015*). This common symbiosis signaling pathway shares a number of genetic components including the machinery to decode the specific calcium signature (*Kistner et al., 2005*) and a few downstream factors such as VPY, a Major Sperm Domain- and ankyrin-repeat-containing protein, that is required for both symbiotic interactions (*Murray et al., 2011*; *Pumplin et al., 2010*). Although some structural features are shared between the AM symbiosis and RNS during intracellular colonization, others are remarkably different. A whole set of those RNS-specific responses is controlled by the master regulator NIN, a transcription factor orchestrating the transcriptional reprogramming of cells required to sustain bacterial accommodation and concomitant cell proliferation to induce the formation of a nodule primordium (*Schauser et al., 1999*; *Liu et al., 2019a*; *Vernié et al., 2015*). This spatial and temporal coordination of infection and nodule organogenesis represents a hallmark of RNS (*Oldroyd and Downie, 2008*).

Intracellular invasion of rhizobia is uniquely enabled by a membrane-confined transcellular tunnel called the IT. In many legumes such as *M. truncatula* and *L. japonicus*, IT formation is initiated from an infection chamber (IC) (*Fournier et al., 2015*) that forms upon curling of a growing root hair and subsequent entrapping of rhizobia (*Callaham and Torrey, 1981*). Repolarization of targeted secretion towards this compartment enables local cell wall modifications (e.g. by secretion of the NPL to the IC *Liu et al., 2019a*) leading to its radial expansion (*Fournier et al., 2015*) and a highly confined local invagination of the plasma membrane, which marks the onset of IT initiation (*Gage, 2004*). The IT then extends through the trichoblast and subsequently progresses transcellular through the root cortex (*Libbenga and Harkes, 1973*). IT maintenance and progression require a highly coordinated interplay between the migrating nucleus, a cytosolic column ahead of the IT, actin, and dense microtubule arrays (*Fournier et al., 2015*; *Perrine-Walker et al., 2014*; *Qiu et al., 2015*; *Timmers et al., 1999*; *Yokota et al., 2009*). IT propagation in underlying cortical tissues is supported by local cell wall modifications (*Su et al., 2023a*) and repolarization of these cells, forming pre-infection thread (PIT) structures (*van Brussel et al., 1992*). This ensures guidance of the symbionts towards the newly divided cortical cells constituting the core of the nodule primordium. There, rhizobia are released from the ITs into the host cell, where they differentiate into nitrogen-fixing bacteroids (*Vasse et al., 1990*).

Over the last decades, reverse and forward genetics have identified several genes required for bacterial infection (*Roy et al., 2020*; *Tsyganova et al., 2021*). Among these is *RPG*, which is specifically expressed upon rhizobial inoculation and encodes a protein of unknown function (*Arrighi et al., 2008*). A loss of RPG in *M. truncatula* results in the development of aberrant ITs and poorly colonized nodules (*Arrighi et al., 2008*). Nuclear localization of RPG when heterologously over-expressed in *Nicotiana benthamiana* leaf epidermal cells suggested a putative role as a transcription factor (*Arrighi et al., 2008*), but its cellular and molecular function in *M. truncatula* has not been understood in detail. The protein, however, recently re-gained great attention, when two independent comparative

phylogenomic studies identified *NFP, NIN,* and *RPG* as the only genes that have been consistently lost in non-nodulating species belonging to the Fagales/Fabales/Cucurbitales/Rosales (FaFaCuRo) clade (*Griesmann et al., 2018*; *van Velzen et al., 2018*). This evolutionary pattern linking these genes and the ability to form RNS makes them prime candidates for engineering symbiotic nitrogen fixation in cereals.

Here, we demonstrate that RPG is an essential component of the infectosome complex (*Liu et al., 2019b*; *Roy et al., 2020*). This multi-protein assembly associates with the very tip of growing ITs and with the nuclear periphery. A loss-of-function mutation in *RPG* entirely blocks the recruitment of VPY to infectosome foci. Furthermore, a highly confined membrane polarity domain at the IT tip, cytoskeleton connectivity between this site, and the nucleus and targeted secretion to the IT are aberrant in a *rpg* mutant. Thus, we propose that RPG provides specificity to a conserved cellular machinery that enables rhizobial accommodation.

## Results

### *RPG* controls the development of infection thread structures in *M. truncatula*

An ethyl methanesulfonate (EMS)-induced mutant of *M. truncatula* accession A17 (hereafter named *rpg-1, Figure 1A*) was previously reported to develop enlarged ITs and poorly colonized nodules (*Arrighi et al., 2008*), indicating that *RPG* is required for rhizobial infection. In order to genetically confirm this function, we isolated a second independent, homozygous mutant allele from the *M. truncatula Tnt1* transposon insertion collection in the R108 accession carrying an insertion at 948 bp downstream of the start codon (NF11990, hereafter named *rpg-2, Figure 1A*). We phenotypically analyzed both mutant alleles in parallel. Confocal imaging of infected root hairs 35 days post-inoculation (dpi) with a fluorescent GFP-tagged strain of the compatible symbiont *S. meliloti* (*S. meliloti*-GFP) revealed similar alterations of IT morphology in both *rpg* mutants, with ITs showing a distinctive bulbous and thick appearance compared to straight and thin ITs of their corresponding wild-types (*Figure 1B*). To better characterize and quantitatively evaluate this peculiar IT phenotype, we scored the diameter of the tube at three different points along the IT (proximal to the infection chamber, intermediate and distal, *Figure 1C*). While the diameter size of ITs developed by the wild-types had little variation (*Figure 1D*), reflecting the tight regulation exerted by the host over the development of this tubular structure, the diameter of ITs formed in roots of the *rpg* mutants was highly variable, and significantly wider at all measure points (*Figure 1D*). This indicates a loss of stringent host-controlled IT maintenance in these mutants.

To exclude the possibility that the IT phenotype is simply due to an altered root hair morphology, we quantified the diameter of differentiated root hair cells in the maturation zone of *rpg* mutant alleles and their corresponding wild-types in un-inoculated conditions (*Figure 1—figure supplement 1A–C*). All four genotypes showed similar root hair diameters (*Figure 1—figure supplement 1C*), whose mean values and variability are in line with a previous study reporting on the diameter of mature root hairs in *M. truncatula* A17 WT (*Shaw et al., 2000*). This confirms a specific role for *RPG* during infection, which is consistent with its previously reported expression pattern (*Arrighi et al., 2008*).

Despite their defective growth, ITs of both *rpg* mutants occasionally penetrated cortical cell layers. To examine the morphology of cortical ITs with cellular resolution, we cleared and imaged roots at 14 dpi with *S. meliloti*-mCherry and stained them with the cell wall stain Calcofluor white to highlight the cell borders. Similar to what we observed in root hairs, ITs within cortical cells appeared consistently enlarged and bulbous in roots of both *rpg* mutants alleles compared to their wild-type genotypes, indicating that *RPG* is required to control IT development in both epidermal and cortical cell layers (*Figure 1—figure supplement 2A*).

We next compared the nodulation capacity of the mutant alleles by quantifying the number and type of nodules formed at 21 dpi with *S. meliloti*-LacZ. We found that nodular structures formed on roots of both *rpg* mutant alleles but their total amount and the number of infected nodules were significantly reduced compared to their corresponding wild-types. In addition, *rpg* mutants failed to develop fully elongated nodules and formed mainly bumps at this time point (*Figure 1—figure supplement 2B*). This further supports the hypothesis that *RPG* is not required to initiate nodule organogenesis but is essential for efficient infection and full colonization. Nodulation defects were

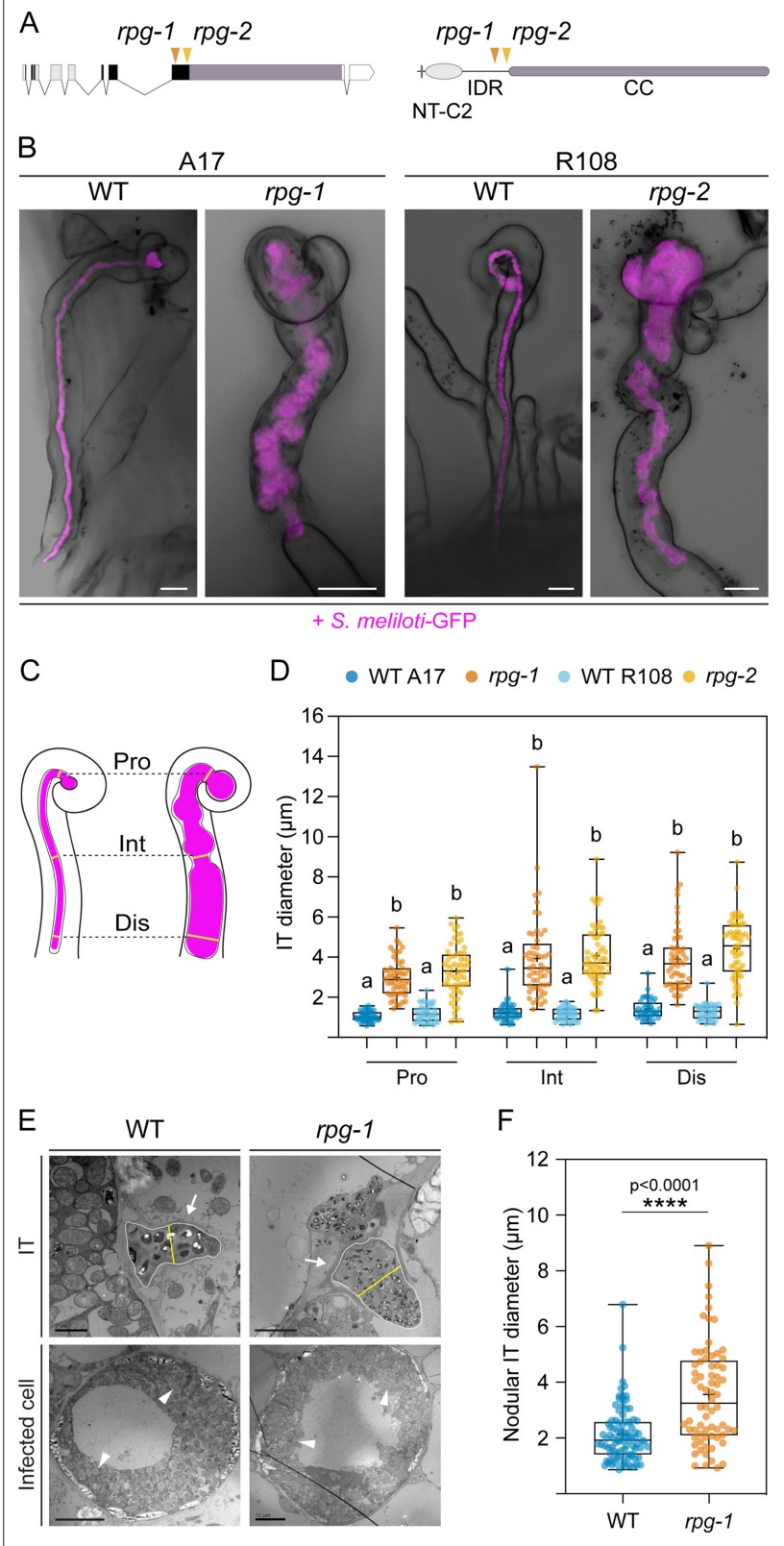

**Figure 1.** *Rhizobium-directed polar growth (RPG) is required for the maintenance of infection thread morphology.*
(**A**) Gene (left) and protein (right) structure of RPG showing the position of the mutations of two different mutant alleles (*rpg-1*, **Arrighi et al., 2008**, orange arrowhead; *rpg-2*, this study, yellow arrowhead). Both mutations map to the third exon of the *RPG* gene, corresponding to the disordered region of the protein located upstream of

*Figure 1 continued on next page*

*Figure 1 continued*

the coiled-coil domain. NT-C2=N-terminal C2 domain; IDR = intrinsically disordered region; CC = coiled coil domain. (**B**) Representative confocal images of aberrant ITs formed 35 dpi within root hairs of *rpg-1* and *rpg-2* roots compared to thin and elongated infection threads (ITs) of the corresponding wild-types (WT). Images are overlaid with intensity projections of fluorescence and bright field channels. *S. meliloti*-GFP is shown in magenta. Scale bars = 10 µm. (**C**) Schematic visualization of the method used to quantify morphological IT defects. The IT diameter was measured on the fluorescent channel at three different points along the IT length: in the proximity of the infection chamber (Pro), in the intermediate part of the IT (Int), and in the distal part of the IT (Dis). (**D**) IT diameters scored on roots of *rpg* mutants and corresponding WTs at the three measured points shown in (**C**). Letters indicate statistically significant differences according to Kruskal-Wallis multiple comparison analysis followed by Dunn's post-hoc test. Data are from two independent experiments with 10 (WT A17); 11 (*rpg-1*); 10 (WT R108); 10 (*rpg-2*) plants analyzed. n=44 (WT A17), 53 (*rpg-1*), 44 (WT R108), 61 (*rpg-2*) ITs. (**E**) Representative transmission electron microscopy (TEM) sections obtained from WT and *rpg-1* nodules showing IT structures (arrows) with a thicker appearance in *rpg-1* compared to WT. The organization of infected cells and symbiosome (arrowheads) morphology are similar in the two genotypes. The yellow line in the upper panel indicates the IT diameter, positioned at the center of the IT area (white outline), perpendicular to its longest axis. Scale bars = 2 µm (left, upper panel), 5 µm (right, upper panel), 10 µm (lower panels). (**F**) Nodular IT diameters were measured on TEM sections from nodules formed on roots of *rpg-1* and WT. Asterisks indicate statistical significance based on a Mann-Whitney test with p-values <0.05 (*), <0.01 (**), <0.001 (***), and <0.0001 (****). Data are from two independent experiments, with six (WT) and five (*rpg-1*) nodules analyzed, each harvested from a different plant. n=35 (WT) and 31 (*rpg-1*) infected cells; n=90 (WT) and 70 (*rpg-1*) nodular ITs. In each box plot, the top and bottom of each box represent the 75th and 25th percentiles, the middle horizontal bar indicates the median and the whiskers represent the range of minimum and maximum values. Crosses represent sample means.

The online version of this article includes the following source data and figure supplement(s) for figure 1:

**Source data 1.** IT diameter and statistical analysis.

**Source data 2.** Nodular IT diameter and statistical analysis.

**Figure supplement 1.** *Rhizobium-directed polar growth (RPG)* is not required to maintain root hair morphology in un-inoculated conditions.

**Figure supplement 1—source data 1.** RH diameter and statistical analysis.

**Figure supplement 2.** Cortical infection threads and nodule phenotype of *rpg* mutants.

**Figure supplement 2—source data 1.** Nodule number and statistical analysis.

---

more pronounced in *rpg-2* than in *rpg-1*, which well correlates with previous findings showing that a mutation within the infection-related gene *LIN*, which is required for IT growth, affects more severely nodule development when occurring in the genetic background R108 compared to A17 (*Guan et al., 2013*).

To assess the relevance of *RPG* for bacterial release inside nodules, we examined ultra-thin sections of infected nodules formed on *rpg-1* and A17 WT via transmission electron microscopy (TEM). TEM micrographs of nodule-infected cells revealed larger IT structures present in nodule tissues of *rpg-1* compared to WT but WT-like cell organization and symbiosome morphology (*Figure 1E*). By measuring the diameter of nodular ITs we found that those formed in *rpg-1* nodules were significantly wider compared to WT (*Figure 1F*). This indicates that normal IT development in nodules requires *RPG*, but bacterial release and differentiation are not directly dependent on this gene.

Altogether these data genetically confirm that *RPG* is essential to sustain efficient bacterial infection in *M. truncatula* and clearly supports a role in controlling proper assembly and progression of infection thread structures in host root tissues.

## The coiled-coil domain of RPG is necessary for infection thread maintenance

To further understand how RPG mediates IT maintenance, we analyzed the molecular determinants required to mediate the functionality of the encoded protein during the infection of root hairs.

*RPG* encodes a 1255 aa long protein with an N-terminal segment containing a putative nuclear localization signal (NLS) and an N-terminal C2 domain (NT-C2) predicted to mediate membrane tethering (*Zhang and Aravind, 2010*), linked by a serine-rich intrinsically disordered region (IDR) to a long alpha-helical C-terminal extension with several predicted coiled-coil (CC) regions possibly mediating

oligomerization (*Figure 2A*). A similar structural organization and high degrees of sequence conservation of the NT-C2 domain and the N-terminal part of the coiled-coil region were found in several homologous proteins (*Arrighi et al., 2008*, *Figure 2—figure supplement 1*). However, since all of these proteins remained uncharacterized at the functional and molecular level, the contribution of the structural features to protein functionality is unknown.

To gain more insights into the structural requirements for RPG functionality, we generated a series of truncation derivatives corresponding to (i) the alpha-helical C-terminal coiled-coil region (aa 331–1255, RPG-CC; *Figure 2A*), (ii) an N-terminal truncated version of the coiled-coil region, lacking the region conserved among RPG homologs (aa 474–1255; RPG-tCC; *Figure 2A*), and (iii) the N-terminal part containing the NLS, the NT-C2 domain and the disordered region (aa 1–330; RPG-NT; *Figure 2A*). To further extend localization pattern analysis, we additionally produced N-terminal mCherry fusions of the full-length protein (RPG-FL; *Figure 2A*) and all truncated variants. Functionality of the constructs was first assessed by testing their capability to genetically complement IT morphology when expressed in *rpg-1* roots under the control of a functional native *RPG* promoter. For this, we used *Agrobacterium rhizogenes*-mediated gene transfer, generating composite plants having stably transformed roots but unaltered shoot genotypes, each representing an independent transformant (*Boisson-Dernier et al., 2001*). Composite plants expressing mCherry-RPG-FL phenocopied the WT-like IT morphology, indicating that N-terminal tagging does not impair protein functionality during IT development within root hairs (*Figure 2B–C*). Interestingly, we found that the sole coiled-coil region of RPG (mCherry-RPG-CC) was equally able to significantly restore the morphological defects of *rpg-1* ITs, even though with lower efficiency when directly compared to RPG-FL (*Figure 2B–C*). This ability was fully dependent on the N-terminal 143 aa of this region, as expression of mCherry-RPG-tCC did not rescue the *rpg-1* IT phenotype (*Figure 2B–C*). These data suggest that major molecular determinants conferring functionality to RPG reside in its coiled-coil domain. This was further supported by the fact that the mCherry-NT variant alone did not restore normal IT growth in the *rpg-1* mutant (*Figure 2B–C*). On the other side, the lower complementation efficacy of mCherry-RPG-CC compared to mCherry-RPG-FL (*Figure 2C*) clearly indicates the relevance of the N-terminal region for full RPG functionality and further suggests that the NLS, the NT-C2 domain, and the disordered region might constitute an important regulatory unit required to fine-tune the protein function or regulate its stability.

## RPG resides in infectosome foci hosting the VPY-LIN-EXO70H4 protein complex

To analyze RPG localization patterns, we performed live-cell imaging of WT *M. truncatula* roots (A17) expressing mCherry-RPG-FL under the control of the native *RPG* promoter. Following rhizobial inoculation, the RPG protein consistently coalesced into a few discrete cytoplasmic puncta that were visible in both uninfected and infected root hairs imaged 4–7 dpi (*Figure 3A*). In uninfected root hairs, such puncta were found in the vicinity of the tip (uRH; *Figure 3A*) while in curled root hairs at the early stage of infection, these puncta were located in close proximity to the IC, and adjacent to the entrapped bacteria (IC; *Figure 3A*). In root hairs harboring growing ITs, RPG-labeled puncta were systematically present in the perinuclear cytoplasm and associated with the tip of the extending IT (IT; *Figure 3A*). Closer inspection showed that IT tip-associated puncta formed a patch juxtaposed to the growing IT tip ahead of progressing bacteria (*Figure 3—figure supplement 1A*). When ITs exhibited multiple branches, RPG-labeled puncta were associated with the tip of each branch (*Figure 3—figure supplement 1B*). Alongside with such punctate patterning, a diffuse RPG signal was also detectable in the nucleus and the cytoplasm of most, though not all, infected root hairs (18/28), but its intensity was orders of magnitudes weaker compared to the signal intensity detected in the puncta (*Figure 3—figure supplement 1C–D*) and thus currently not accessible for functional analysis. Altogether, these data indicate that RPG preferentially accumulates in cytoplasmic puncta associated with infection sites during bacterial accommodation within root hairs (*Figure 3B*). This is in contrast to the exclusive nuclear localization previously reported for RPG when being ectopically expressed in *N. benthamiana* leaves (*Arrighi et al., 2008*). While we also observed nuclear accumulation of RPG upon constitutive, heterologous over-expression in *N. benthamiana* leaves, we additionally observed RPG signal diffuse in the cytoplasm, accumulating in the nucleolus and in a few small cytoplasmic punctate structures associated to the perinuclear area but also found in other areas of pavement cells

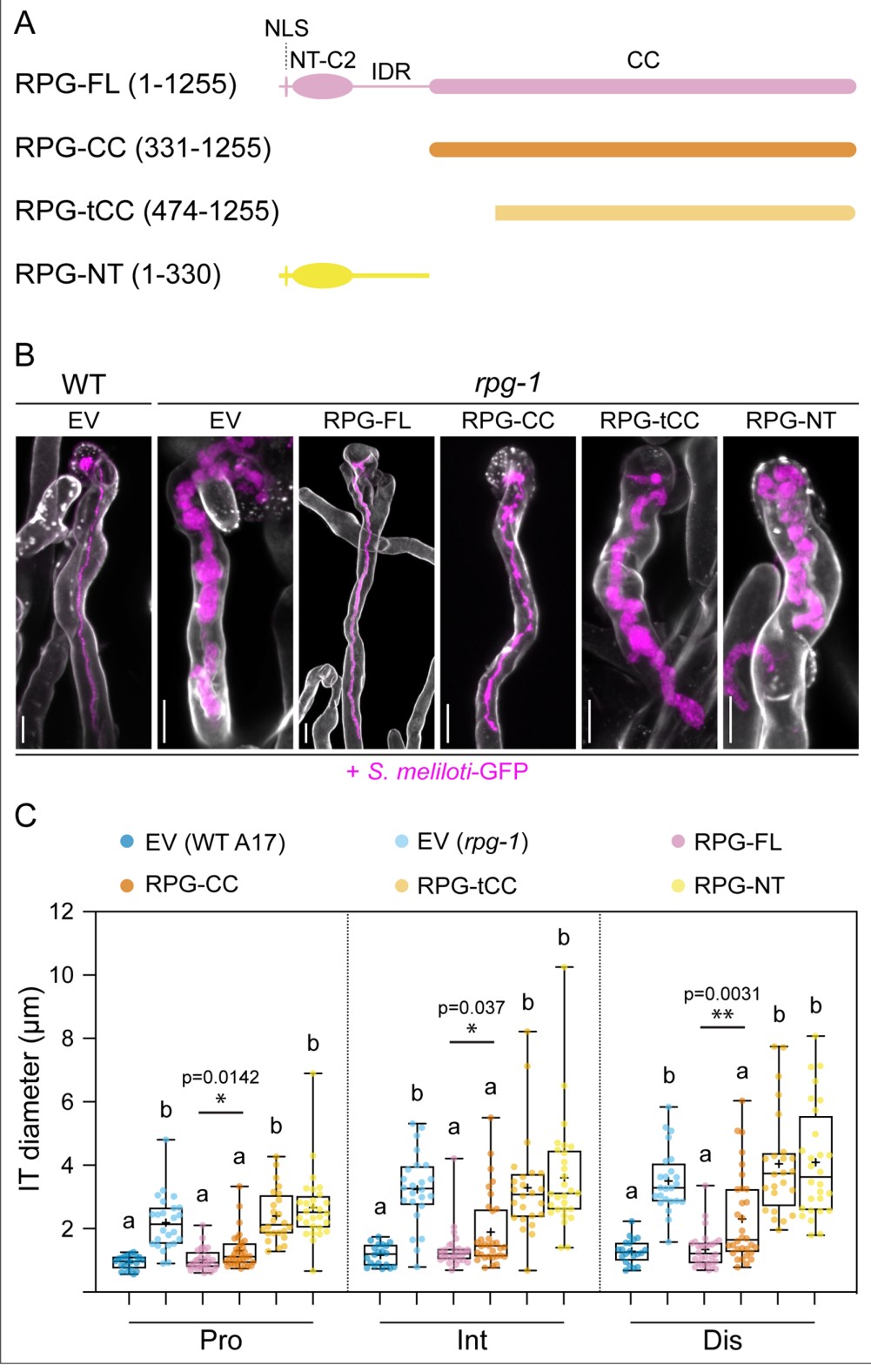

**Figure 2.** The coiled-coil domain of Rhizobium-directed polar growth (RPG) is necessary to restore WT-like infection thread morphology in *rpg-1* root hairs. (**A**) Schematic representation of the RPG full-length protein and deletion derivatives used for complementation assays. Numbers indicate amino acids included in each fragment. NLS = nuclear localization signal; NT-C2=N-terminal C2 domain; IDR = intrinsically disordered region; CC = coiled coil domain. (**B**) Representative confocal images of infection threads (ITs) developed within root

*Figure 2 continued on next page*

*Figure 2 continued*

hairs of wild-type (WT) and *rpg-1* transgenic roots expressing the different constructs 21 dpi with *S. meliloti*-GFP (magenta). Cell walls were stained with Calcofluor white (white). Images are merges of maximum-intensity projections of fluorescent channels. EV = empty vector. Scale bars = 10 µm. (**C**) IT diameters scored at the three different points schematically represented in *Figure 1C*. In the box plot, the top and bottom of each box represents the 75th and 25th percentiles, the middle horizontal bars indicate the median and the whiskers represent the range of minimum and maximum values. Crosses represent sample means. Letters indicate statistically significant differences according to Kruskal-Wallis multiple comparison analysis followed by a Dunn's post-hoc test. A Mann-Whitney test was performed to compare RPG-FL and RPG-CC with p-values <0.05 (*), <0.01 (**), and <0.001 (***). Data are from one of two independent experiments showing similar tendencies, with five (EV in WT), six (EV in *rpg-1*), seven (RPG-FL,), eight (RPG-CC), six (RPG-tCC), seven (RPG-NT) composite plants analyzed (one root per plant) and four ITs per roots. n=20 (EV in WT), 24 (EV in *rpg-1*), 28 (RPG-FL), 32 (RPG-CC), 24 (RPG-tCC), 28 (RPG-NT) ITs.

The online version of this article includes the following source data and figure supplement(s) for figure 2:

**Source data 1.** IT diameter and statistical analysis.

**Figure supplement 1.** Sequence conservation among Rhizobium-directed polar growth (RPG) homologs.

---

(*Figure 3—figure supplement 2*). To facilitate the detection of these puncta, we co-expressed RPG with the viral silencing suppressor p19 (*Voinnet et al., 2003*), which reduces post-transcriptional gene silencing impeding gene expression during Agrobacterium-mediated gene transfer (*Johansen and Carrington, 2001*; *Voinnet et al., 2003*). This enhanced the accumulation of RPG in punctate structures, although it concurrently led to the formation of large accumulations of the fusion protein in the perinuclear area (*Figure 3—figure supplement 2*), most likely caused by the high level of transgene expression reached under co-infiltration conditions (*Jay et al., 2023*). Thus, the coalescence of RPG in cytoplasmic puncta occurs also in *N. benthamiana* heterologous system, as similarly recently reported by a second independent study (*Li et al., 2023*).

To evaluate the contribution of RPG protein domains to its punctate localization, we analyzed the subcellular patterning of the RPG deletion derivatives expressed under the control of the native *RPG* promoter in roots of composite *rpg-1 M. truncatula* plants. When expressed in this mutant background, mCherry-RPG-FL exhibited a comparable localization pattern to what was observed in the WT background (*Figure 3—figure supplement 3A*). We found that both mCherry-RPG-CC and mCherry-RPG-tCC accumulated to bright punctate structures as mCherry-RPG-FL, while the mCherry-RPG-NT signal remained diffuse in the cytoplasm and preferentially accumulated in the nucleus of transformed root hairs (*Figure 3—figure supplement 3B–D*). These results indicate that the coiled-coil domain is necessary and sufficient for RPG localization to discrete puncta (*Figure 3C*) and that the deletion of amino acid residues 331–473, although abolishing its functionality, does not affect its localization to these foci (*Figure 2A–C*, and *Figure 3—figure supplement 3B–C*). In addition, the consistent nuclear accumulation of mCherry-RPG-NT compared to the diffuse nucleo-cytoplasmic localization weakly detectable for RPG-FL, suggests that the coiled-coil domain determines the preferential retention of the protein in cytoplasmic puncta (*Figure 3—figure supplement 3A and D*, *Figure 3—figure supplement 4A–D* and *Figure 3C*). It should, however, be noted that the accumulation of mCherry-RPG-CC at IT-tip associated puncta (*Figure 3—figure supplement 3B* - inset) occurred less frequently (7/22 root hairs harboring infection threads) compared to mCherry-RPG-FL (17/21 root hairs harboring infection threads), while both consistently labeled perinuclear or cytoplasmic puncta. This suggests that optimal targeting and/or accumulation of RPG to the IT tip relies on the presence of its N-terminal part (*Figure 3C*), which well correlates with the reduced capacity of RPG-CC to restore IT morphology in *rpg-1* transgenic roots in comparison to RPG-FL (*Figure 2C*).

The subcellular patterning of RPG resembles the recently described localization of the so-called 'infectosome' complex formed by VPY, LIN, and EXO70H4. These proteins systematically co-localize and interact in IT tip- and perinuclear-associated cytoplasmic puncta during root hair infection (*Liu et al., 2019b*; *Roy et al., 2020*). Thus, we tested whether RPG is a component of this protein assembly. For this, we first co-expressed mCherry-RPG and VPY-GFP under the control of their respective native promoters in roots of composite *M. truncatula* WT plants and imaged them at 4–7 dpi with *S. meliloti*-mCherry. Since the signal from co-expressed RPG- and VPY- fluorophore fusions was consistently weak, to improve its detection and to unambiguously distinguish it from

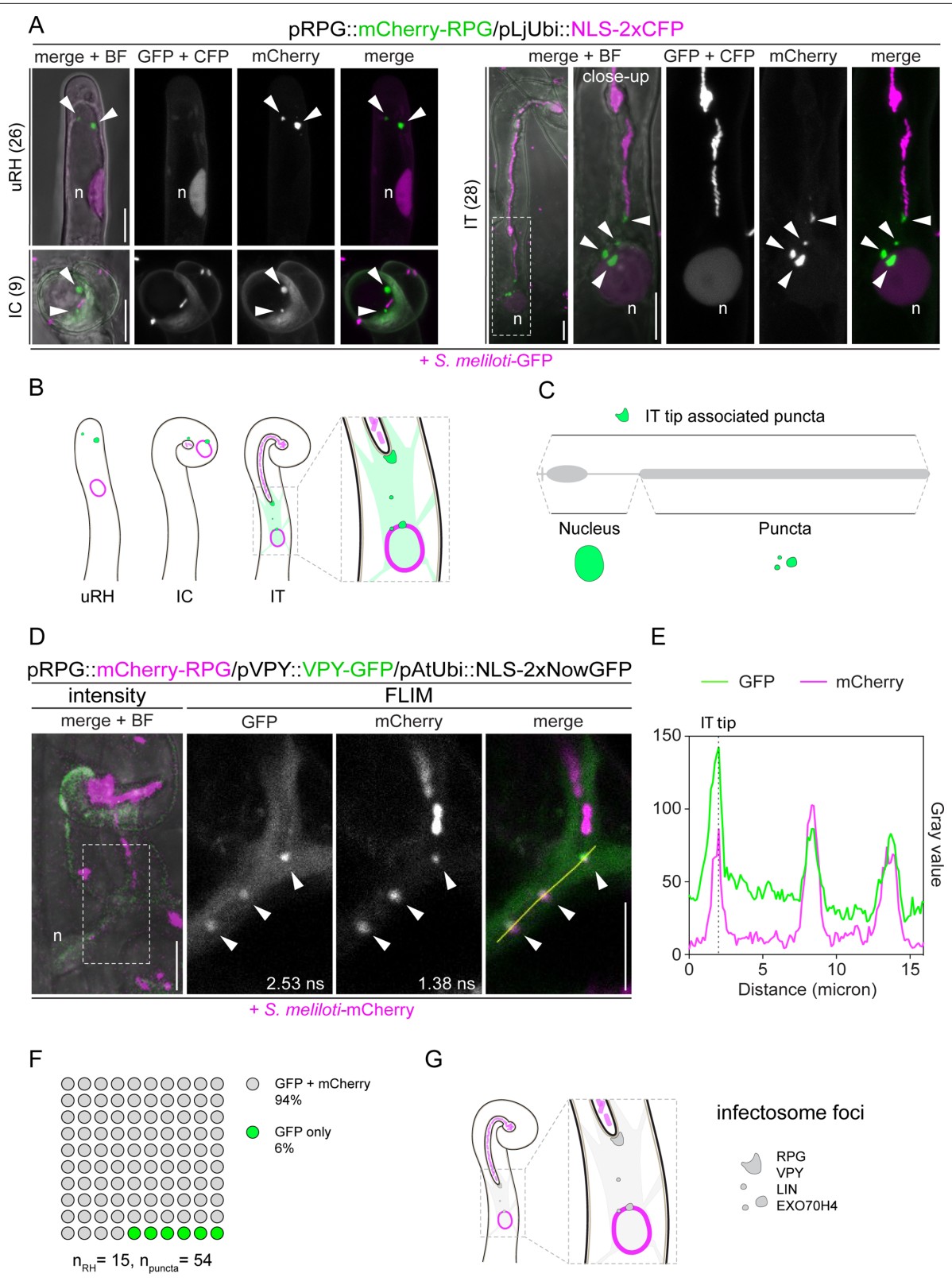

**Figure 3.** Rhizobium-directed polar growth (RPG) is a key component of the VPY-LIN-EXO70H4 infectosome complex. (**A**) Live-cell confocal images showing mCherry-RPG localization in root hairs of WT transgenic roots at 4–7 dpi with *S. meliloti*-GFP. mCherry-RPG accumulates in cytoplasmic punctate structures (arrowheads) which are found close to the apex of uninfected root hairs (uRH), in the vicinity of rhizobia entrapped within the infection chamber of curled root hairs (IC) and associated to the perinuclear cytoplasm and the tip of infection threads (IT). The dashed white line box

*Figure 3 continued on next page*

*Figure 3 continued*

indicates the region shown in the close-up, which is a projection of Z-stacks acquired at higher magnification. A nuclear-localized tandem CFP was used as a transformation marker. Images are maximum intensity projections. Individual channels are false colored in gray. When channels are merged, the GFP/CFP channel is shown in magenta and the mCherry channel is shown in green. The bright field (BF) is overlaid with merged fluorescent channels. The number of cells analyzed per each infection stage is reported at the left side of each panel. At least seven composite plants from three independent experiments were analyzed. (**B–C**) Schematic representations of the subcellular patterning of RPG (green irregular round shapes and diffuse green coloring) in root hairs upon inoculation with rhizobia (light magenta) (**B**) and contribution of RPG protein (gray) domains to its localization pattern (**C**). The nucleus is represented as an oval outlined in magenta (**B**) or colored in green (**C**). (**D**) Co-localization of mCherry-RPG with VPY-GFP in cytoplasmic and IT tip-associated puncta (arrowheads) in root hairs of wild-type (WT) transgenic roots imaged at 4–7 dpi with *S. meliloti*-mCherry. The intensity-based image is a merge of maximum intensity projections of GFP (green) and mCherry (magenta) channels overlaid with the BF channel. The dashed white line box indicates the region shown in Fluorescence Lifetime Imaging Microscopy (FLIM) images. FLIM images are maximum intensity projections of two focal planes (20 repetitions, threshold = 10 photons). Lifetime values (ns) obtained from the exponential reconvolution of decay profiles of GFP and mCherry channels are reported. Individual channels are shown in gray. When components are merged, GFP is shown in green and mCherry in magenta. A nuclear-localized tandem GFP was used as a transformation marker (not visible in these images - see *Figure 3—figure supplement 5*). (**E**) Intensity profile of GFP and mCherry signal along the yellow transect is shown in (**D**). The intensity peaks at the infection thread tip (IT tip) are marked by a dashed black line. (**F**) Quantification of the co-localization in infected root hairs (n=15) displaying the discrete signal from VPY-GFP and/or mCherry-RPG in punctate structures (n=54). Six composite plants from two independent replicas were analyzed. (**G**) Schematic representation of the co-localization of RPG with the infectosome components VAPYRIN (VPY) and LUMPY INFECTION (LIN) in distinct subcellular foci (gray irregular round shapes). The nucleus is represented as a magenta-outlined oval. Rhizobia within infection threads are colored in light magenta. n=nucleus. Scale bars = 10 µm.

The online version of this article includes the following source data and figure supplement(s) for figure 3:

**Source data 1.** FLIM fit values, intensity profile values, and percentage of co-localization.

**Figure supplement 1.** Rhizobium-directed polar growth (RPG) sub-cellular localization pattern in root hairs.

**Figure supplement 1—source data 1.** Intensity profile values.

**Figure supplement 2.** Subcellular localization of Rhizobium-directed polar growth (RPG) in *N. benthamiana* epidermal cells.

**Figure supplement 3.** The coiled-coil region is necessary and sufficient to drive Rhizobium-directed polar growth (RPG) localization to puncta.

**Figure supplement 4.** The coiled-coil domain determines the preferential retention of Rhizobium-directed polar growth (RPG) to cytoplasmic puncta.

**Figure supplement 4—source data 1.** Intensity profile values.

**Figure supplement 5.** Rhizobium-directed polar growth (RPG) co-localizes with VAPYRIN (VPY) during root hair infection.

**Figure supplement 5—source data 1.** FLIM fit values and intensity profile values.

**Figure supplement 6.** Rhizobium-directed polar growth (RPG) co-localizes with LUMPY INFECTION (LIN) during root hair infection.

**Figure supplement 6—source data 1.** FLIM fit values, intensity profile values, and percentage of co-localization.

**Figure supplement 7.** Rhizobium-directed polar growth (RPG) resides in a complex with the infectosome component EXO70H4.

**Figure supplement 7—source data 1.** Raw images of Western Blot analysis.

background fluorescence, we applied Fluorescent Lifetime Imaging Microscopy (FLIM). By performing multiple cycles of excitation and detection of photon arrival times, this method increases the intensity of fluorescence emission. Furthermore, it allows to analyze its decay profile, thereby enabling an increased spatial resolution of genetically encoded fluorophores according to their particular lifetimes (GFP ~2.5–2.7 ns; mCherry ~1.4–1.5 ns, *Sarkisyan et al., 2015*; *Seefeldt et al., 2008*). It additionally allows the subtraction of the autofluorescence signal, having similar spectral characteristics but significantly shorter lifetimes (see materials and methods for more details). We found that mCherry-RPG co-localized with VPY-GFP in punctate structures located at the tip of the IT and in the cytoplasm connecting the nucleus to the IT tip of infected root hairs (*Figure 3D–F* and *Figure 3—figure supplement 5A–C*). A VPY-GFP signal of lower intensity was also diffusely visible in the cytoplasm and nucleus, as previously reported (*Liu et al., 2019b*), coinciding with a weak signal from RPG (*Figure 3D–E* and *Figure 3—figure supplement 5A–C*). Co-expression of GFP-RPG with mCherry-LIN, the latter being transcriptionally driven by the constitutive *Ubiquitin* promoter from *L. japonicus* (pLjUbi), also resulted in distinct co-localization of the two proteins in root hairs undergoing infection (*Figure 3—figure supplement 6A–C*). In both cases, FLIM enabled a clear lifetime-based signal discrimination of the nuclear-localized transformation markers (NLS-2xNowGFP, lifetime ~4.0 ns, *George Abraham et al., 2015*; NLS-3xmScarlet, lifetime ~3.5–3.9 ns, *Bindels et al., 2017*; *Bindels et al., 2020*) from the signal of VPY-GFP and mCherry-LIN, respectively (*Figure 3—figure supplement 5A* and *Figure 3—figure supplement 6A*).

Furthermore, we performed co-immunoprecipitation (Co-IP) assays on protoplasts isolated from Agrobacterium-infiltrated *N. benthamiana* leaves constitutively co-expressing RPG, VPY, and EXO70H4 in different combinations of pairs. When using anti-RFP nanobody traps, GFP-RPG and mCherry-RPG co-purified with mCherry-EXO70H4 and VPY-GFP, respectively (*Figure 3—figure supplement 7A*). To reciprocally validate these data, we performed these assays using anti-GFP nanobody traps. While GFP-RPG consistently co-precipitated with mCherry-EXO70H4 (*Figure 3—figure supplement 7B*), we were unable to pull sufficient amounts of VPY-GFP (*Figure 3—figure supplement 7C*), suggesting that the majority of the VPY protein pool is not associated with RPG under these conditions. Since we experienced the same difficulties as recently reported (*Li et al., 2023*) when trying to purify recombinant RPG from *Escherichia coli* cells, we are not able at this stage to biochemically address the interaction of RPG, VPY, and the other infectosome components. However, given the peculiar coalescence of all of these proteins in cytoplasmic puncta (*Liu et al., 2019b* and this study; *Figure 3G*), we decided to further genetically evaluate the impact of *RPG* on infectosome foci formation in *M. truncatula*, focusing on VPY as one of its central components.

## RPG is necessary and sufficient for VPY recruitment to infectosome foci

Since the characteristic punctate localization of VPY is triggered by rhizobial inoculation and does not depend on functional LIN (*Liu et al., 2019b*), we hypothesized that VPY accumulation to infectosome foci may be triggered by and dependent on RPG. This is further supported by the fact that *RPG* expression is induced in root hairs starting from 2 days following inoculation with *S. meliloti* (*Arrighi et al., 2008*; *Liu et al., 2019a*). To test this hypothesis, we expressed VPY-GFP in roots of WT and *rpg-1* composite plants and visualized its spatial patterning in root hairs at 4–7 dpi with *S. meliloti*-mCherry. Indeed, while we systematically observed bright VPY-positive puncta in infected (18/19 in a total number of eight plants) and uninfected (6/8 in a total number of six plants) root hairs of transformed WT roots, a reliable signal of such punctate subcellular structures in *rpg-1* transformed root hairs was neither detected in infected (19/19 events in a total number of 10 plants) nor in uninfected (8/8 events in a total number of seven plants) root hairs. In all cases, VPY exhibited a diffuse cytoplasmic signal (*Figure 4A* and *Figure 4—figure supplement 1A*). In a few instances (5/19 infection events found in 2 plants out of 10), we observed VPY accumulating close to bacteria entrapped within a curl, seemingly surrounding the site of membrane invagination occurring at the early stage of IT emergence, but no cytoplasmic or perinuclear-associated puncta were visible (*Figure 4—figure supplement 1B*). This is in sharp contrast to what we observed in WT root hairs at a similar stage of infection (*Figure 4—figure supplement 1B*). From these data, we concluded that the symbiont-induced re-localization of VPY to the IT-tip and perinuclear-associated puncta is RPG-dependent (*Figure 4B*).

Next, we asked whether RPG is not only required but also sufficient to drive VPY localization to puncta. To test this, we compared the sub-cellular localization pattern of VPY-GFP in root hairs when ectopically expressed either alone or in combination with mCherry-RPG in roots of WT *M. truncatula* composite plants under un-inoculated conditions. In the absence of RPG, VPY displayed a diffuse cytoplasmic and nuclear signal in all root hairs imaged (40/40 root hairs in a total number of 16 plants, *Figure 4C*). Such sub-cellular patterning was similar to the localization of VPY in root hairs when expressed under its native promoter either in WT transgenic *M. truncatula* roots under un-inoculated conditions (*Liu et al., 2019b*) or in *rpg-1* transgenic roots following inoculation (*Figure 4—figure supplement 1A*). In contrast, constitutive co-expression with mCherry-RPG resulted in VPY coalescing into few punctate structures, mostly found at the perinuclear cytoplasm of root hairs, where it co-localized with RPG (43/52 root hairs in a total number of 18 plants, *Figure 4C–E*). This indicates that RPG alone is sufficient for the efficient recruitment of VPY to subcellular puncta in a symbiont-independent manner (*Figure 4F*).

Altogether, these data point to RPG as a key component of the infectosome machinery controlling IT polar growth, being necessary to modulate rhizobial-induced focusing of VPY to IT tip and perinuclear-associated puncta during this process. The predicted function of the infectosome and the loss of VPY recruitment to IT-tip associated puncta in *rpg-1,* which fails to develop thin and elongated infection threads, strongly suggest that RPG is required to sustain localized polarity at this specific symbiotic domain.

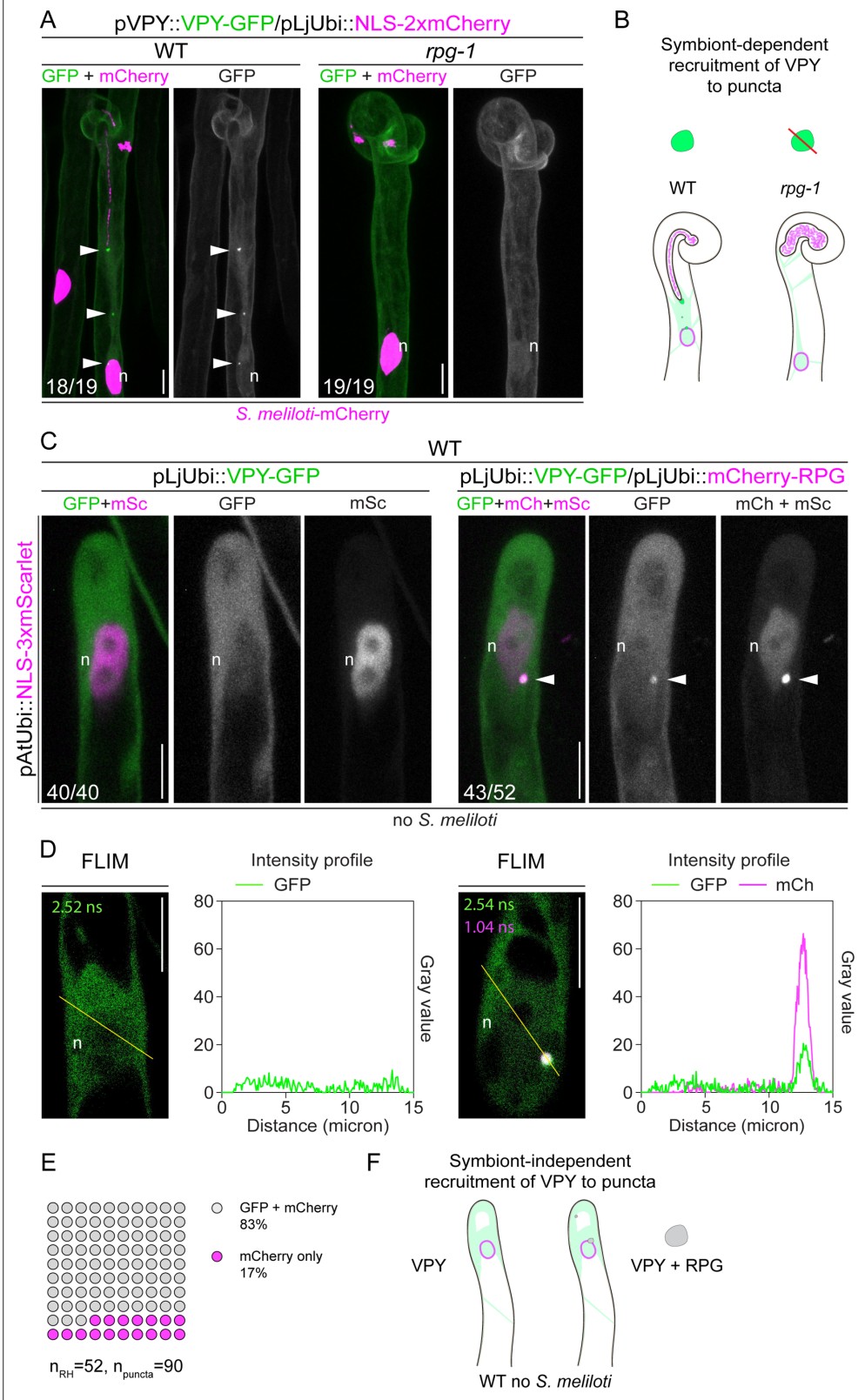

**Figure 4.** Rhizobium-directed polar growth (RPG) is necessary and sufficient for VAPYRIN (VPY) recruitment to infectosome foci. (**A**) Localization pattern of VPY-GFP in root hairs of WT and *rpg-1* transgenic roots imaged at 4–7 dpi with *S. meliloti*-mCherry. VPY-positive puncta (arrowheads) are systematically present in wild-type (WT) but not in *rpg-1* infected root hairs. Numbers indicate frequencies of observations, made on a total number of eight (WT)

*Figure 4 continued on next page*

*Figure 4 continued*

and 10 (*rpg-1*) composite plants. Two independent replicas were performed. A nuclear-localized tandem mCherry was used as a transformation marker. Images are maximum intensity projections. The GFP channel is shown in gray when isolated, in green when merged with mCherry (magenta). (**B**) Schematic representation illustrating the dependency of VPY recruitment to infectosome foci by RPG. Upon infection with *S. meliloti* (light magenta), infectosome foci (green irregular round shape) visible in root hairs of WT transgenic roots are instead undetectable (red diagonal line) in root hairs of *rpg-1*, where a diffuse signal from VPY (light green coloring) labels the cytoplasm and nucleus (magenta outlined oval). (**C**) Subcellular patterning of ectopically expressed VPY-GFP either alone (left panel) or in combination with mCherry-RPG (right panel) in root hairs of WT transgenic roots in the absence of *S. meliloti*. VPY-GFP and mCherry-RPG co-localize in cytoplasmic puncta (arrowheads) which are detectable upon co-expression of the two transgenes, while coalescence of VPY-GFP in such puncta is not observed upon expression of the VPY transgene alone. Images are maximum intensity projections. Individual channels are shown in gray; in merged images, the GFP channel is shown in green and the mCherry/mScarlet channel is shown in magenta. A nuclear-localized triple mScarlet driven by the *Arabidopsis Ubiquitin* promoter was used as a transformation marker. Images are representative of 40 (VPY-GFP) and 52 (VPY-GFP/mCherry-RPG) root hairs from a minimum of 16 composite plants from two independent replicas. Numbers indicate frequencies of observations. (**D**) Fluorescence Lifetime Imaging Microscopy (FLIM) images (single focal planes, 20 repetitions, one photon threshold) of root hairs shown in (**C**) and intensity profiles along the yellow transects drawn on the respective images. Lifetime values (ns) obtained from the exponential reconvolution of decay profiles of each channel are reported. (**E**) Quantification of the co-localization in root hairs (n=52) displaying discrete signal from VPY-GFP and/or mCherry-RPG in punctate structures (n=90). 18 composite plants from two independent replicas were analyzed. (**F**) Schematic illustration showing the symbiont-independent recruitment of VPY to cytoplasmic puncta co-labeled by RPG (gray irregular round shape) upon ectopic co-expression of the two transgenes in root hairs of un-inoculated WT roots. The nucleus is represented as a magenta-outlined oval. The diffuse cytoplasmic signal from VPY is illustrated with light green coloring. n=nucleus. Scale bars = 10 µm.

The online version of this article includes the following source data and figure supplement(s) for figure 4:

**Source data 1.** FLIM fit values, intensity profile values, and percentage of co-localization.

**Figure supplement 1.** VAPYRIN (VPY) recruitment to puncta is dependent on Rhizobium-directed polar growth (RPG).

## RPG is required to maintain PI(4,5)P$_2$ enrichment on the membrane of the IT tip

In tip-growing plant cells such as root hairs and pollen tubes, confined polarity at the apical membrane domain is sustained by the local enrichment of low abundant phosphoinositide species such as PI(4,5)P$_2$, which serve as landmarks to recruit diverse protein effectors regulating cytoskeleton rearrangements, vesicle trafficking and signaling (*Ischebeck et al., 2010*; *Noack and Jaillais, 2020*). As the polarized IT tip likely recruits part of the machinery controlling polarity establishment during root hair growth (*Gage, 2004*), we hypothesized that local enrichment of PI(4,5)P$_2$ occurs at the site of symbiont accommodation. To test this, we expressed the well-characterized genetically encoded PI(4,5)P$_2$ biosensor mCitrine-2xPH$^{PLC}$ (*Simon et al., 2014*) under the control of the infection-induced *ENOD11* promoter (*Boisson-Dernier et al., 2005*; *Journet et al., 2001*) in roots of WT *M. truncatula* composite plants and analyzed its localization patterns following inoculation with *S. meliloti*-mCherry (4–8 dpi). To increase the robustness of our analysis, we adopted a systematic FLIM-based approach that allowed us (i) to unambiguously identify local membrane enrichment of the mCitrine-PH$^{PLC}$ probe (Lifetime ~3 ns; *Söhnel et al., 2016*) and to discriminate it from cell wall autofluorescence (Lifetime ≤1 ns) and (ii) to integrate lifetime measurements obtained from the observation of multiple events in a visually-intuitive form using the Phasor approach (*Malacrida et al., 2021*; *Ranjit et al., 2018*) and to statistically compare them (*Figure 5—figure supplement 1A*; see materials and methods for full description).

In uninfected growing root hairs, PI(4,5)P$_2$ enriched at the apical membrane domain (*Figure 5—figure supplement 2A* and *Figure 5—figure supplement 3A*) while PI(4,5)P$_2$ was locally and specifically enriched at the membrane surrounding invading bacteria emerging from the IC at the onset of infection (*Figure 5A* – IC and *Figure 5—figure supplement 3B*). This indicates that symbiont accommodation coincides with membrane polarization at this host-microbe interface. In root hairs harboring actively growing ITs, we consistently observed local enrichment of the phosphoinositide on the membrane surrounding the tip and the terminal part of the extending tube, connected to

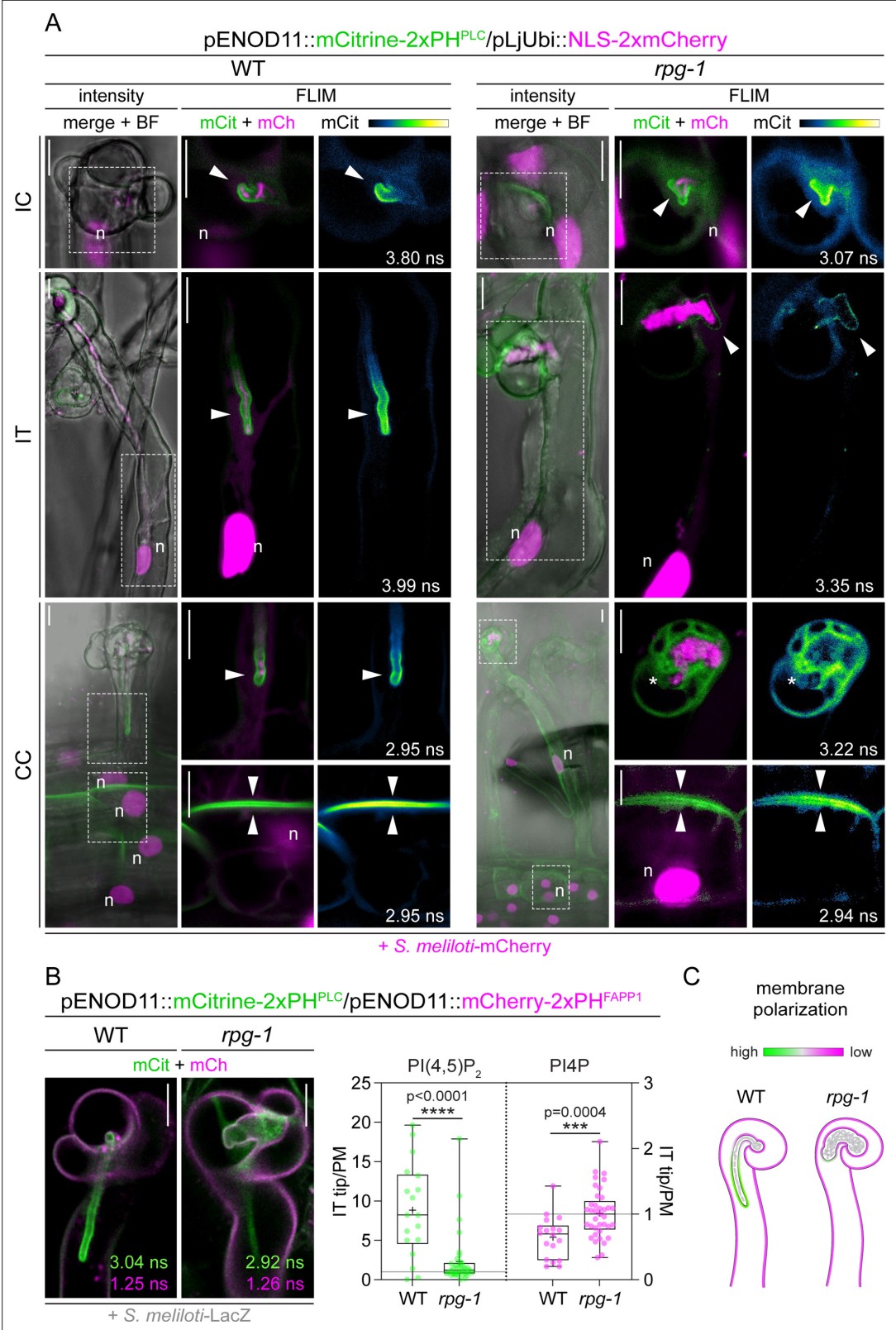

**Figure 5.** Rhizobium-directed polar growth (RPG) is required to maintain strong membrane polarization at the infection thread (IT) tip. (**A**) Live-cell confocal images of root hairs from transgenic WT and *rpg-1* roots expressing the PI(4,5)P$_2$ biosensor mCitrine-2xPH$^{PLC}$ at 4–8 dpi with *S. meliloti*-mCherry. The distribution of mCitrine-2xPH$^{PLC}$ in WT and *rpg-1* transgenic roots are compared during the emergence of rhizobia from the infection chamber (IC), in root hairs hosting infection threads (IT) and in cortical cells underlying epidermal infection events (CC). Membrane domains where

*Figure 5 continued on next page*

*Figure 5 continued*

differential enrichment of PI(4,5)P$_2$ occurred are indicated with arrowheads. Cytoplasmic signal of the biosensor in *rpg-1* infected root hairs is indicated with an asterisk. Intensity-based images are merges of maximum intensity projections of mCitrine (green) and mCherry (magenta) channels overlaid with the bright field (BF) channel. The dashed white line box in intensity images indicates the region shown in Fluorescent Lifetime Imaging Microscopy (FLIM) images. FLIM images are single focal planes; the mCitrine component is shown in green when merged with the mCherry component (magenta) and in Green Fire Blue when isolated, with yellow indicating the maximum intensity and blue a low level of fluorescence. Lifetime values (ns) obtained from the exponential reconvolution of decay profiles of the mCitrine channel are reported. A nuclear-localized tandem mCherry was used as a transformation marker. Each infection stage has been monitored on a minimum of three composite plants from at least three independent replicas. n=nucleus. (**B**) Co-visualization of PI4P and PI(4,5)P$_2$ in root hairs harboring ITs from wild-type (WT) and *rpg-1* transgenic plants co-expressing mCitrine-2xPH$^{PLC}$ and mCherry-2xPH$^{FAPP1}$ at 4–7 dpi with *S. meliloti*-LacZ. Images are merges of mCitrine (green) and mCherry (magenta) components with indicated lifetime obtained from exponential reconvolution of decay profiles of the respective channels. A nuclear-localized triple mScarlet was used as a transformation marker (not visible in these images). The IT tip versus plasma membrane fluorescence ratio calculated for each biosensor in WT and *rpg-1* infected root hairs is reported in the box plot, where the top and bottom of each box represents the 75th and 25th percentiles, the middle horizontal bar indicates the median and whiskers represent the range of minimum and maximum values. Crosses represent sample means. Horizontal gray lines are positioned at y=1. Asterisks indicate statistical significance based on a Mann-Whitney test with p-values <0.05 (*), <0.01 (**), <0.001 (***), and <0.0001 (****). Data are from three independent experiments with 13 (WT) and 14 (*rpg-1*) composite plants analyzed. n=18 (WT) and 37 (*rpg-1*) infected root hairs. (**C**) Schematic representation of the differential enrichment of PI(4,5)P$_2$ (green) and PI4P (magenta) on membranes of WT and *rpg-1* infected root hairs, indicative of a high and low membrane polarization characterizing the tip of WT and *rpg-1* ITs, respectively. Bacteria within infection threads are colored in gray. Scale bars in (**A**) and (**B**) = 10 μm.

The online version of this article includes the following source data and figure supplement(s) for figure 5:

**Source data 1.** FLIM fit values, IT tip/PM values, and statistical analysis.

**Figure supplement 1.** A Fluorescent Lifetime Imaging Microscopy (FLIM)-based systematic approach was adopted to visualize PI(4,5)P$_2$ membrane enrichment.

**Figure supplement 1—source data 1.** FLIM fit values.

**Figure supplement 2.** PI(4,5)P$_2$ patterning in transgenic roots of wild-type (WT) and *rpg-1*.

**Figure supplement 2—source data 1.** FLIM fit values.

**Figure supplement 3.** Phasor analysis statistically supports loss of infection thread (IT) membrane polarization occurring in *rpg-1*.

**Figure supplement 3—source data 1.** FLIM fit values and Phasor analysis values.

**Figure supplement 4.** A mutant PI(4,5)P$_2$ biosensor does not accumulate at membrane domains during rhizobial infection.

**Figure supplement 4—source data 1.** FLIM fit values.

**Figure supplement 5.** PI(4,5)P$_2$ patterning in transgenic roots of *rpg-1*.

**Figure supplement 5—source data 1.** FLIM fit values.

**Figure supplement 6.** Co-visualization of PI4P and PI(4,5)P$_2$ in wild-type (WT) and *rpg-1* infected root hairs.

the nucleus by a dense column of cytoplasm (*Figure 5* – IT and *Figure 5—figure supplement 3C*), confirming that bacterial progression is associated with strong polarization of the IT membrane. Notably, we observed that local PI(4,5)P$_2$ enrichment also occurred on the basal membrane of root hair cells hosting polarized ITs extending within a cytoplasmic column connected to the root hair cell base, and on the apical membrane of underlying cortical cells (*Figure 5A* – CC, *Figure 5—figure supplement 2B–C* and *Figure 5—figure supplement 3D*). This fully coincided with the reorganization of the cortical cell cytoplasm into a dense central column occupied by the nucleus, which is typical of PIT responses (*van Brussel et al., 1992*; *Figure 5A* - CC and *Figure 5—figure supplement 2B–C*). This suggests that local membrane polarization occurs on the trajectory of IT propagation and might serve to direct local changes in the cell wall and apoplastic space in preparation for the transcellular passage of the bacteria.

To further verify that the observed membrane localization of the 2xPH$^{PLC}$ biosensor reflects PI(4,5)P$_2$ enrichment, we generated a mutant version of the probe bearing amino acid substitutions R37D and R40D, which substantially reduce its PI(4,5)P$_2$-binding capacity and abolish its interaction with the plasma membrane (mCitrine-2xPH$^{PLC-mut}$; *Yagisawa et al., 1998*; *Ivanov and Harrison, 2019*). Expression of this mutated probe under the control of the *ENOD11* promoter in WT transgenic roots resulted in exclusively diffuse cytoplasmic signal in both uninfected root hairs (*Figure 5—figure supplement 4* - uRH), infected root hairs (*Figure 5—figure supplement 4* – IC and IT) and in cortical cells underlying primary infection events (*Figure 5—figure supplement 4* - CC) when assessed at 4–7 dpi with *S. meliloti*-mCherry.

Next, we evaluated the impact of a loss of RPG on infection-dependent membrane repolarization by expressing the PI(4,5)P$_2$ probe in *rpg-1* transgenic roots and monitoring its distribution pattern upon inoculation (*S. meliloti* mCherry, 4–8 dpi). Here, the apex of uninfected growing root hairs was labeled by the PI(4,5)P$_2$ biosensor as in the WT (*Figure 5—figure supplement 2A* and *Figure 5—figure supplement 3A*). PI(4,5)P$_2$ patterning was also similar to WT at the very initial stage of bacterial accommodation, where we observed a local enrichment occurring on the membrane surrounding the bacteria protruding from the IC (*Figure 5A* – IC and *Figure 5—figure supplement 3B*). Thus, RPG is not required to establish membrane polarization at this stage. However, PI(4,5)P$_2$ failed to strongly enrich on the majority of membranes (21/33 ITs from 10 plants) of typically enlarged infection threads developed by the *rpg-1* mutant. Here, the signal of the probe remained either low and uniformly labeling the membrane, cytoplasmic, or almost absent (*Figure 5A*, *Figure 5—figure supplement 3C* and *Figure 5—figure supplement 5A*), indicating that functional RPG is required to maintain IT-confined PI(4,5)P$_2$ polarization during symbiont accommodation. When present, the enrichment was often found on short membrane protrusions formed at multiple polar sites on the enlarged infection structures developed within *rpg-1* root hairs, further supporting that not the establishment but rather the maintenance and confinement of symbiont-induced polarization is RPG-dependent (*Figure 5—figure supplement 3C* and *Figure 5—figure supplement 5A*). In addition, we found that the development of polar domains at the base of infected root hair cells and on the apical membrane of underlying cortical cells occurred in *rpg-1* (*Figure 5A* – CC, *Figure 5—figure supplements 3D and 5B*), implying that RPG is not necessary for these pre-infection responses but exclusively for IT maintenance and progression. This is in agreement with previous findings showing that PIT formation and microtubule rearrangements in cortical cells of *rpg-1* are not perturbed (*Arrighi et al., 2008*). However, such responses were uncoupled from IT polarization and nuclear-guided progression of the tube within the root hair (*Figure 5A*-CC and *Figure 5—figure supplement 5B*), suggesting that functional RPG is important to maintain synchronization between cells during the infection process.

To provide statistical validation of our analysis, we compared the Phasor fingerprints of fluorescence emission (see Materials and methods for a detailed description) at different membrane domains analyzed in multiple cells of different plants for each genotype as an indicator of PI(4,5)P$_2$ enrichment (*Figure 5—figure supplement 3E*). This analysis revealed a statistical difference in the Phasor position of fluorescence emission at the IT membrane but not at other domains, with the consistent shift towards the left part of the Phasor plot observed for *rpg-1* ITs being indicative of a low PI(4,5)P$_2$ membrane enrichment (*Figure 5—figure supplement 3E*).

To further characterize and compare the membrane identity of WT and *rpg-1* ITs, we co-visualized the distribution of PI(4,5)P$_2$ together with its synthetic precursor PI4P, which is highly enriched at the plasma membrane of plant cells and contributes to define its identity (*Simon et al., 2016*). For this, we co-expressed mCitrine-2xPH$^{PLC}$ with the PI4P biosensor mCherry-2xPH$^{FAPP1}$ (*Simon et al., 2014*) both under the control of the *ENOD11* promoter in roots of WT and *rpg-1* composite plants. To avoid that the intense signal from fluorescently-labeled bacterial symbionts interferes with the visualization of any of the two biosensors, we inoculated composite plants with a non-fluorescent rhizobial strain (*S. meliloti*-LacZ).

In WT infected root hairs, the tip-polarized enrichment of the PI(4,5)P$_2$ biosensor on the IT membrane spatially coincided with low enrichment of the PI4P probe, which instead preferentially accumulated on the root hair plasma membrane (*Figure 5B* and *Figure 5—figure supplement 6*). Such differential membrane enrichment of the two biosensors was much reduced in most of *rpg-1* infected root hairs, where PI4P was similarly enriched on the root hair plasma membrane and on the infection thread membrane, the latter being characterized by low accumulation of the PI(4,5)P$_2$ probe (*Figure 5B* and *Figure 5—figure supplement 6*). We quantitatively confirmed these observations by assessing the IT tip versus plasma membrane fluorescence ratio of each biosensor in infected root hairs, statistically validating the differential enrichment of the two phosphoinositides in the membrane surrounding the tip of WT and *rpg-1* ITs (*Figure 5B*). These results support our previous set of data, indicating a role for RPG in sustaining localized membrane polarization at the tip of the IT (*Figure 5C*).

Since PI(4,5)P$_2$-enriched membrane domains constitute important platforms steering cytoskeleton rearrangements and targeted vesicle delivery (*Bloch et al., 2016*; *Doumane et al., 2021*; *Stanislas et al., 2018*; *Synek et al., 2021*) and as a last set of experiments to verify the role of RPG in

maintaining localized polarity at the IT tip, we decided to test how microtubule organization and polar secretion associated to this symbiotic domain are affected in *rpg-1*.

## Tip-to-nucleus microtubule connectivity is perturbed in *rpg-1*

During IT growth, a dense array of endoplasmic microtubules (EMTs) connects the IT tip to the nucleus migrating towards the root hair base and maintenance of such functional connection is thought to be necessary to guide IT progression within host cells (*Timmers et al., 1999*; *Perrine-Walker et al., 2014*; *Fournier et al., 2008*; *Fahraeus, 1957*).

Since nuclear movement appeared uncoupled from IT growth in *rpg-1* (*Figure 5A* - CC), we tested if the tip-to-nucleus connectivity is perturbed in *rpg-1* by comparing microtubule organization in WT and *rpg-1* root hairs harboring ITs using live-cell imaging. To reduce possible perturbations of microtubule dynamics given by the ectopic expression of exogenous tubulin isoforms (*Celler et al., 2016*), we labeled microtubules expressing an N-terminal GFP fusion of an endogenous tubulin beta isoform (*Medtr8g107250,* hereafter SYMTUB) up-regulated in infected root hairs (*Breakspear et al., 2014*) under the control of a native 1.6 kb promoter (pSYMTUB::eGFP-SYMTUB) in WT and *rpg-1* transgenic roots. In most of the WT root hairs hosting ITs imaged at 4–8 dpi with *S. meliloti*-mCherry (11/14 ITs from six plants), the nucleus and the IT were connected by EMTs extending from the nucleus and forming a cone-shaped array surrounding the IT tip, which was oriented towards the root hair base, following the nuclear trajectory (*Figure 6A* and *Figure 6—figure supplement 1*). This patterning is consistent with previous reports (*Timmers et al., 1999*; *Perrine-Walker et al., 2014*), supporting the validity of our marker. When imaging *rpg-1* infected root hairs, such connection appeared loosened or completely absent in the majority of cases (31/48 ITs from 14 plants). Here, EMTs mostly appeared dispersed or stretched, no longer bridging the tip and the nucleus (*Figure 6A* and *Figure 6—figure supplement 1*), suggesting that loss of nuclear-mediated IT guidance occurs in *rpg-1*. In addition, the tip of ITs formed within *rpg-1* infected root hairs appeared frequently misoriented with respect to the direction of growth (19/48 ITs), pointing towards the root hair shank instead of following the repositioning of the nucleus down the root hair (*Figure 6A* and *Figure 6—figure supplement 1*), further hinting for a defect in controlling IT directionality in this mutant.

To further quantitatively evaluate and compare the tip-to-nucleus connectivity in WT and *rpg-1* infected root hairs, we measured the distance between the nucleus and the IT tip (*Figure 6B* and *Figure 6—figure supplement 1*). Such parameter was reported to vary between 0–40 µm in *M. truncatula* WT root hairs hosting actively growing ITs, with an average value of 20±10 µm (*Fournier et al., 2008*). In good agreement with this previous analysis, most WT-infected root hairs exhibited a similar tip-to-nucleus distance in our experiments (*Figure 6B*, median value of 32, 16 µm). A consistent increase in distancing between the nucleus and the IT tip was observed in infected root hairs where EMTs were no longer tightly connecting these two poles (2/3 ITs with no clear microtubule-based connection, *Figure 6—figure supplement 1*). The tip-to-nucleus distance measured in infected root hairs of *rpg-1* exhibited a higher variability compared to WT, with a major proportion of root hairs displaying a greater distance separating the nucleus from the IT tip (*Figure 6B*, median value of 63, 68 µm), coinciding in most of the cases with the loss of a microtubule-based connection between these two domains (23/31 ITs, *Figure 6—figure supplement 1*). Though less frequently when compared to WT, we did observe EMTs linking the IT tip to the nucleus in *rpg-1* infected root hairs, maintaining a tip-to-nucleus distance comparable to what was observed in WT (*Figure 6—figure supplement 1*), suggesting that the formation of a physical connection between these two poles per se does not depend on *RPG*. Rather, the loss of RPG functionality perturbs the maintenance of such connection and of an optimal distancing between the nucleus and the extending tip during IT growth (*Figure 6C*).

## Polar secretion of NPL requires functional RPG

The cell wall modifying enzyme NPL, which is essential for IT initiation and elongation in both *L. japonicus* and *M. truncatula* (*Xie et al., 2012*; *Liu et al., 2019a*) is secreted to the IT apoplast and accumulates particularly at IT tip regions in *M. truncatula* (*Liu et al., 2019a*; *Su et al., 2023a*), strongly suggesting that its tip-targeted secretion is required for IT polar extension. To evaluate the role of RPG in NPL targeting, we compared the localization pattern of this enzyme in WT and *rpg-1* infected root hairs. For this, we expressed an NPL-GFP fusion under the control of the native *NPL* promoter in WT and *rpg-1* roots and imaged infected root hairs at 4–8 dpi. In WT root hairs harboring growing ITs

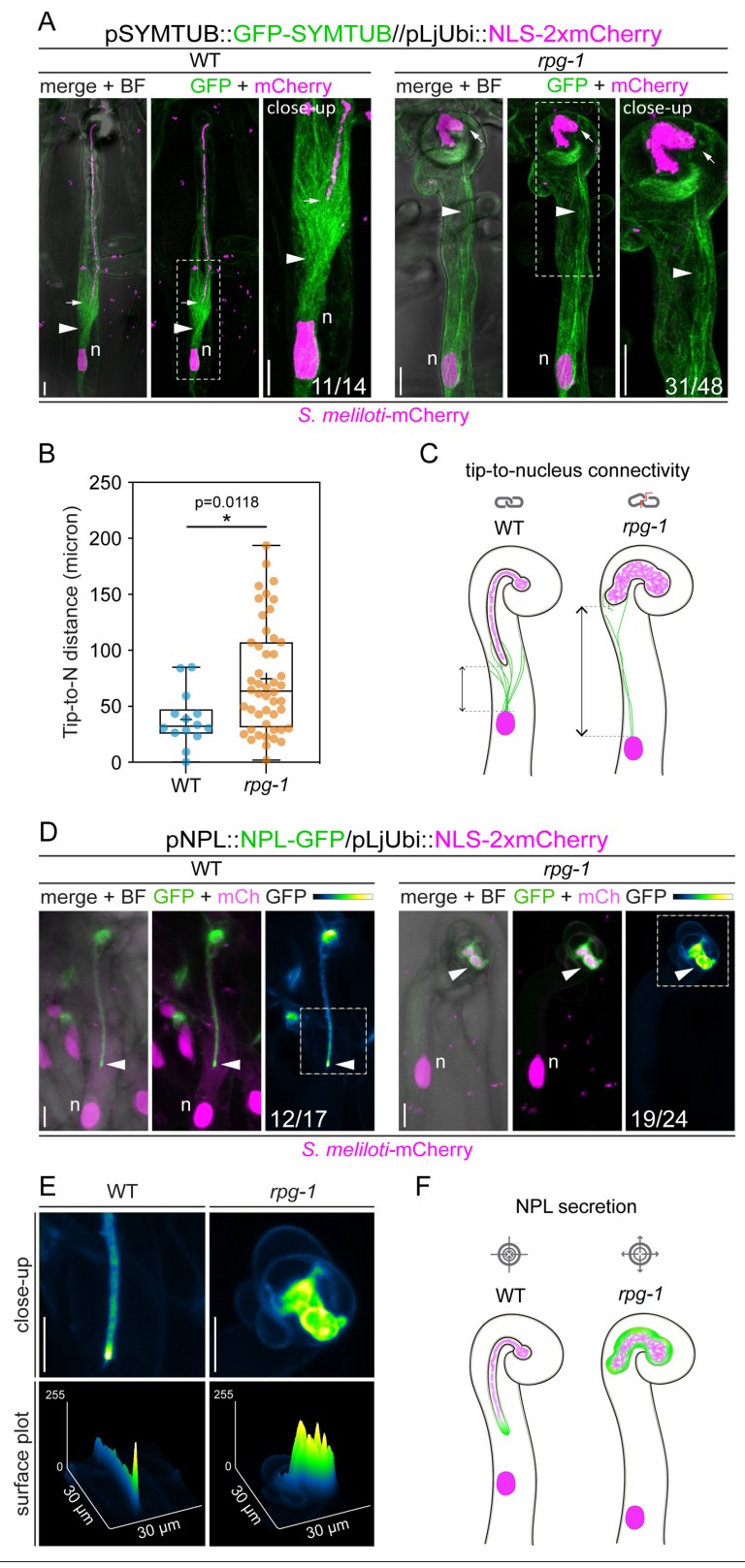

**Figure 6.** The tip-to-nucleus microtubule connectivity and polar secretion of Nodule pectate lyase (NPL) are affected in *rpg-1*. (**A**) In vivo imaging of microtubule patterning in root hairs hosting infection threads from wild-type (WT) and *rpg-1* transgenic roots expressing GFP-SYMTUB at 4–8 dpi with *S. meliloti*-mCherry. The endoplasmic microtubule array (arrowhead) bridging the nucleus (n, magenta) and the infection thread (IT) tip

*Figure 6 continued on next page*

*Figure 6 continued*

(arrow) in WT root hairs appears dispersed and stretched in *rpg-1* root hairs, where the IT tip does not follow the path of the migrating nucleus. A nuclear-localized tandem mCherry was used as a transformation marker. Images are merges of maximum intensity projections of GFP (green) and mCherry (magenta) channels after deconvolution of the corresponding stack series using the Huygens Professional software (Scientific Volume Imaging, The Netherlands). The bright field (BF) channel is overlaid with the merge. The dashed white line boxes indicate the region shown in the close-up. Numbers indicate frequencies of observation made on a total number of six (WT) and 14 (*rpg-1*) composite plants from two independent replicas. (**B**) Distance between the nucleus (**N**) and the IT tip measured in infected root hairs of WT and *rpg-1* transgenic roots expressing GFP-SYMTUB. In the box plot, the top and bottom of each box represent the 75th and 25th percentiles, the middle horizontal bar indicates the median and the whiskers represent the range of minimum and maximum values. Crosses represent sample means. Asterisks indicate statistical significance based on a Mann-Whitney test with p-values <0.05 (*), <0.01 (**), <0.001 (***), and <0.0001 (****). n=14 (WT) and 48 (*rpg-1*) infected root hairs. (**C**) Schematic representation of infected root hairs showing endoplasmic microtubules (green curved lines) maintaining the nucleus (magenta oval) and the IT tip at a close distance (double-arrow line) in WT. Such tip-to-nucleus connectivity (gray icon) is altered (gray icon with red line) in *rpg-1* infected root hairs, where microtubules appear rarefied and do not bridge the IT tip and the nucleus, being separated by an increased distance. Dashed light gray lines indicate the position of the nucleus and the tip. Bacteria within infection threads are colored in light magenta. (**D**) Live-cell confocal images showing focal accumulations of NPL-GFP at the IT tip (arrowhead) in WT root hairs in contrast to its unrestricted distribution in the apoplastic space surrounding ITs developed within root hairs of *rpg-1* transgenic roots. A nuclear-localized tandem mCherry was used as a transformation marker. Images are maximum intensity projections. The GFP channel is shown in green when merged with the mCherry channel (magenta), or in Green Fire Blue when isolated, with yellow indicating the maximum intensity and blue a low level of fluorescence. The bright field (BF) channel is overlaid with the merge. Numbers indicate frequencies of observations made on a total number of 10 (WT) and nine (*rpg-1*) composite plants from two independent replicas. (**E**) Close-up and surface plot of the regions bounded by the dashed white line boxes in (**D**). The signal of NPL-GFP in WT ITs is described by a major intensity peak positioned on the advancing tip compared to multiple intensity peaks detected on ITs of *rpg-1*. (**F**) Schematic representation of infected root hairs showing the accumulation of NPL (green) restricted to the apoplastic space surrounding the tip of WT ITs in contrast to its accumulation in a broader apoplastic domain in IT of *rpg-1*, indicating that targeted secretion of this enzyme (gray icon with arrows pointing to center) is not correctly maintained in this mutant (gray icon with arrows pointing outside). Bacteria within infection threads are colored in light magenta. Scale bars = 10 μm.

The online version of this article includes the following source data and figure supplement(s) for figure 6:

**Source data 1.** Tip-to-nucleus distance and statistical analysis.

**Figure supplement 1.** Microtubule patterning in wild-type (WT) and *rpg-1* infected root hairs.

---

and in agreement with a previous report (***Liu et al., 2019a***), NPL consistently showed a preferential accumulation in the apoplastic space surrounding the tip when compared to the older parts of the IT (***Figure 6B–C***). By contrast, NPL accumulated in the whole lumen of the enlarged ITs without any increased accumulation at a specific site in most *rpg-1* root hairs (***Figure 6D–E***). This indicates that RPG is required to restrict the secretion domain of NPL during infection (***Figure 6F***).

## Discussion

The capacity to intracellularly accommodate nitrogen-fixing bacteria is a hallmark of the RNS. Bacterial internalization requires profound restructuring of the host cell to form an infection thread compartment via polarized secretion of plasma membrane and cell wall material to a single focal point. Spatially and temporally coordinated polar progression of IT structures determines the establishment of a functional nitrogen-fixing association.

Here, we demonstrate that RPG is a crucial modulator of the infectosome machinery sustaining the polar growth of intracellular ITs. RPG coalesces with VPY and LIN in IT tip- and perinuclear-associated puncta (infectosome foci, ***Figure 3D–G***, ***Figure 3—figure supplement 6A–C***) and is indispensable for VPY recruitment into these cytoplasmic structures during IT progression (***Figure 4A–B*** and ***Figure 4—figure supplement 1A–B***). A cooperative function of RPG with the infectosome complex in regulating IT polar growth is consistent with the phenotypes of *rpg*, *vpy*, and *lin* mutants, exhibiting strong defects in the establishment and/or the maintenance of IT progression but not being affected in nodule organogenesis (***Kiss et al., 2009***; ***Murray et al., 2011***; ***Liu et al., 2019b***; ***Li et al., 2023***, ***Figure 1B–D***

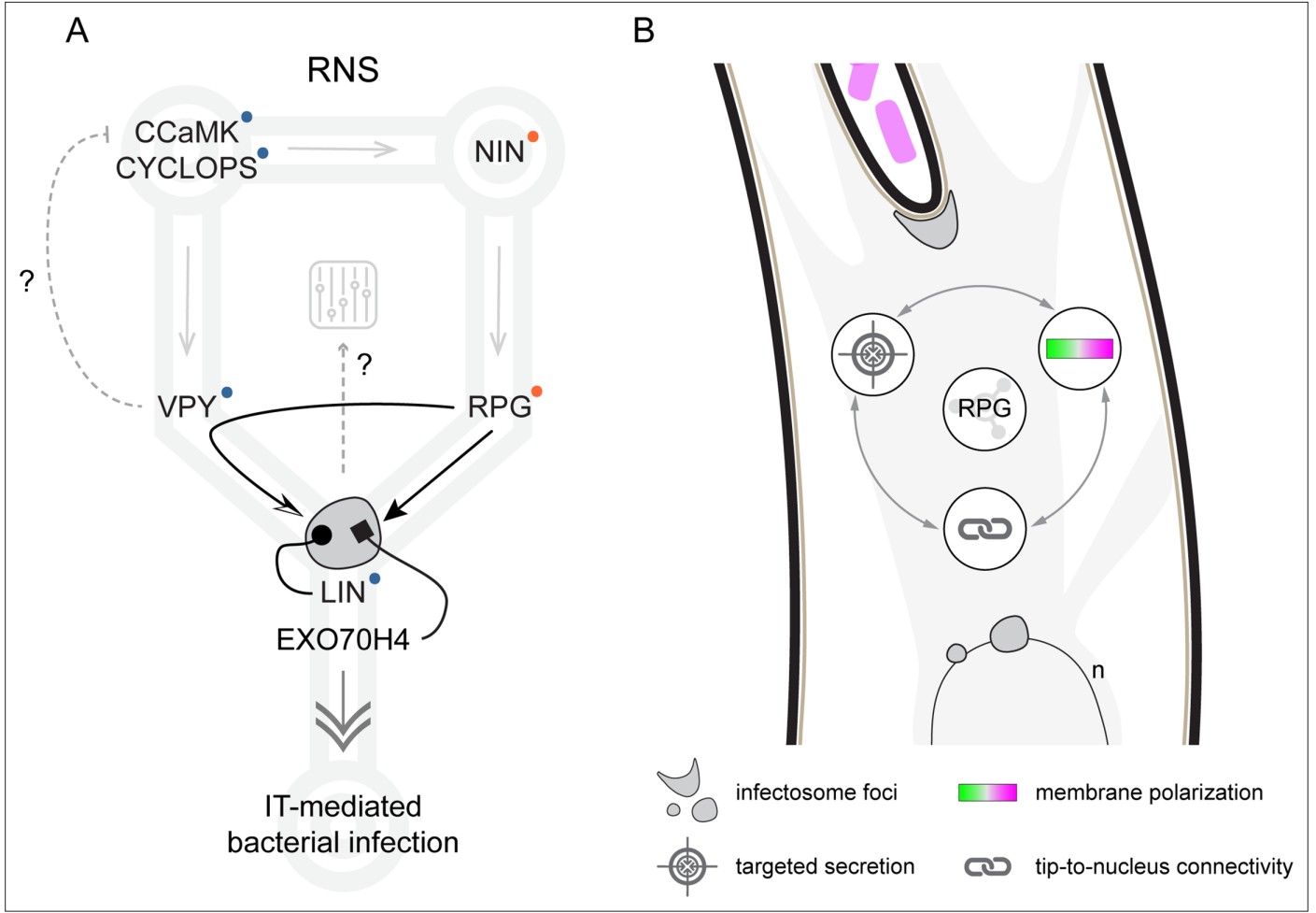

**Figure 7.** Proposed model for Rhizobium-directed polar growth (RPG) role during intracellular rhizobial infection. (**A**) During Root Nodule Symbiosis (RNS), NIN-dependent transcription of *RPG* (gray arrow, *Liu et al., 2019a*; *Soyano and Hayashi, 2014*) leads to accumulation of the encoded protein (black arrow) within infectosome foci hosting VAPYRIN (VPY), LUMPY INFECTION (LIN), and EXO70H4 (gray irregular round shape, this study, *Liu et al., 2019a*). The coalescence of VPY into cytoplasmic foci is fully dependent on RPG (curved arrow, this study), while LIN is required to stabilize VPY (dotted arrow, *Liu et al., 2019a*, *Liu et al., 2021*). Since VPY and LIN belong to a conserved endosymbiotic signaling pathway including the core components CCaMK and CYCLOPS (*Radhakrishnan et al., 2020*, blue dots), we propose that RPG serves as a specificity determinant for infectosome-related endo- and exocytotic events (double arrow) enabling IT-mediated rhizobial infection. This is further supported by the capacity of RPG to interact with the exocyst subunit EXO70H4 (rectangular arrow). RPG-dependent segregation of VPY in infectosome foci might additionally allow to modulate the symbiotic signaling circuit (gray dashed arrow), if the putative negative regulation exerted by VPY on CCaMK (dashed gray flat arrow) during AM (*Lindsay et al., 2022*) would be conserved during RNS. Gray arrows indicate transcriptional activation/dependency (according to *Murray et al., 2011*; *Singh et al., 2014*; *Liu et al., 2019a*; *Li et al., 2023*). Orange dots indicate loss of the corresponding gene in non-nodulating species of the FaFaCuRo clade. (**B**) RPG represents a protein hub for infectosome assembly and its function is required to maintain a functional interplay between membrane polarization, tip-to-nucleus connectivity, and targeted secretion enabling infection thread (IT) polar progression.

and *Figure 1—figure supplement 2A–B*). Moreover, the peculiar enlargement of ITs developed by *rpg* mutants, indicative of a loss of unidirectional polarized growth, is well explained by the failure of *rpg-1* to maintain the typical VPY punctate localization observed in WT root hairs (*Figure 4A–B* and *Figure 4—figure supplement 1A–B*). Interestingly, infectosome-like and VPY-positive compartments have also been observed during the AM symbiosis (*Pumplin et al., 2010*; *Zhang et al., 2015*) and *vpy* mutants exhibit strong defects in this association (*Feddermann et al., 2010*; *Pumplin et al., 2010*). Functionally, the accumulation of VPY in such compartments seems more important for full arbuscule development rather than for controlling initial fungal infections (*Lindsay et al., 2022*). In contrast, the correlation between loss of VPY-recruitment into infectosome foci in the *rpg* mutant and its strong infection defects (*Figure 1B–C* and *Figure 4A–B*) argues for a higher relevance, and perhaps extended function, of these cytoplasmic structures during bacterial accommodation. In line

with this, we hypothesize that RPG serves as a specificity determinant for infectosome-related endo- and exocytotic events during RNS (*Figure 7A*). This is further supported by the transcriptional dependency of *RPG* from NIN (*Liu et al., 2019a*; *Soyano and Hayashi, 2014*) and is in line with *RPG* being one of the few essential factors that have been lost in non-nodulating species of the FaFaCuRo clade (*Griesmann et al., 2018*; *van Velzen et al., 2018*). Upon recruitment of VPY into the infectosome complex, it is further stabilized by LIN (*Liu et al., 2019b*). This can also be assumed for CERBERUS, the ortholog of LIN in *L. japonicus*, which has been shown to stabilize VPY protein levels (*Liu et al., 2021*) and to interact with RPG in this closely related legume (*Li et al., 2023*). Different from *RPG*, the genes encoding for VPY and LIN are retained in angiosperm species forming intracellular symbioses, together with those encoding for the central components of the common symbiosis signaling pathway CCaMK and CYCLOPS, which are essential for both AM and RNS (*Radhakrishnan et al., 2020*). Intriguingly, VPY was recently proposed to coordinate signaling and cellular accommodation during AM by negatively regulating the nuclear-localized symbiosis signaling kinase CCaMK via direct interaction (*Lindsay et al., 2022*). If such regulatory function would be conserved during rhizobial infection, the ability of RPG to sequester VPY to cytoplasmic foci might additionally represent a way to modulate the input/output of a conserved endosymbiotic signaling circuit to adequately mediate bacterial infection (*Figure 7A*).

While the existence of infectosomes is supported by this and other studies (*Liu et al., 2019b*; *Zhang et al., 2015*), the full nature of this presumably membrane-less compartment remains to be investigated further. It is, however, tempting to draw parallels to the so-called 'polarisome,' a macro-molecular complex coordinating the polarized growth of yeasts and fungal hyphae (*Xie and Miao, 2021*). Polarisomes are dense spot-like structures localized at the site of polarized growth and are formed by a number of core components and many cooperative interactors (*Xie and Miao, 2021*). The polarisome is also required to maintain the Spitzenkörper (*Crampin et al., 2005*), a vesicle supply center located at the tip during hyphal growth, whose positioning is correlated with growth directionality (*Riquelme and Sánchez-León, 2014*). Interestingly, a myosin protein has recently been shown to mediate focal packing of the polarisome (*Dünkler et al., 2021*). The structural similarity of the RPG C-terminal extension with myosins, as supported by structural predictions and as previously suggested (*Arrighi et al., 2008*), and the RPG-dependent recruitment of VPY into discrete cytoplasmic foci (*Figure 4A–F*), further supports the hypothesis that RPG functions as an organizational hub protein for infectosome assembly and polarity maintenance during intracellular rhizobial infections (*Figure 7B*). It should, however, be considered that a vesicle-associated fraction of RPG, which is compatible with the predicted lipid binding capability of its NT-C2 domain (*Zhang and Aravind, 2010*), as well as of the other infectosome components outside the puncta might contribute to their functionalities. Although the diffuse cytoplasmic signal exhibited by RPG, VPY, LIN, and EXO70H4 (*Figure 3—figure supplement 3C–D*; *Liu et al., 2019b*) might represent a such a fraction, it cannot be reliably investigated at present due to the vesicle size and/or fluorescence intensity. Within the RPG protein, the core functional scaffolding element is the coiled-coil domain as it is sufficient to mediate the punctate localization of RPG and to complement IT progression in root hairs (*Figure 2A–C* and *Figure 3—figure supplement 3A-D*). The latter is unlikely to be determined by transcript stability as all deletion variants were consistently detected as fluorescent proteins when performing live-cell imaging (*Figure 3—figure supplement 3A-D*).

Rendering the infectosome complex dysfunctional in *rpg* mutants results in the loss of the highly polarized membrane enrichment of $PI(4,5)P_2$ at the IT tip (*Figure 5A–C*) and unconfined secretion of key factors such as NPL (*Figure 6D–F*) required for IT progression to the growing IT tip (*Liu et al., 2019a*; *Xie et al., 2012*). As in many eukaryotic polarized systems, where localized exocytosis targets $PI(4,5)P_2$-rich membrane domains, accumulation of this phosphoinositide was also shown to occur on perimicrobial membranes during pathogenic infections (*Qin et al., 2020*) and during AM symbiosis, where $PI(4,5)P_2$ specifically accumulated at the tips of intracellular linear hyphae (*Ivanov and Harrison, 2019*). In eukaryotes, the generation of $PI(4,5)P_2$ involves the phosphorylation of position 5 of PI4P catalyzed by phosphatidylinositol-4-phosphate 5-kinases (PIP5K) localized at the plasma membrane and in the nucleus (*Colin and Jaillais, 2020*). At least ten genes encoding for isoforms of this enzyme are contained in the Medicago genome and, interestingly, two of them are induced during infection thread development within root hairs (*Medtr3g073100*, *Medtr7g089950*; *Liu et al., 2019a*), one of which (*Medtr7g089950*) in a NIN-dependent manner (*Liu et al., 2019a*). This further supports our

data and suggests that increased PI(4,5)P$_2$ accumulation during infection might occur, at least in part, via transcriptional upregulation of its biosynthetic enzymes. However, the trafficking/activity of these and/or other PI(4,5)P$_2$-metabolizing enzymes are also likely to contribute to the observed membrane enrichment, which is possibly a result of complex feedback loops occurring between those enzymes and their protein effectors such as, for instance, small GTPases, as reported for other systems (*Di Paolo and De Camilli, 2006*). In line with this, a ROP (Rho of a plant) GTPase and its GEF effector were shown to be required for infection thread growth in *L. japonicus* (*Liu et al., 2020*), and expression of at least one gene encoding for a RopGEF is induced during IC and IT formation in Medicago (*Breakspear et al., 2014*). Ultimately, the generation of a stably polarized cell arises from the functional interplay between membrane-associated polarity circuits and consequent downstream cellular reorganization (*Ramalho et al., 2022*), with localized exocytosis being central to the process of tip growth, as shown for pollen tube growth guidance in *Arabidopsis* (*Luo et al., 2017*). It is noteworthy that maintenance of the IC and secretion of NPL into the symbiotic apoplastic compartment is unaltered in *rpg* mutants, but remains unconfined (*Figure 6D–F*). The capacity of RPG to interact with EXO70H4 (*Figure 3—figure supplement 7*) residing within infectosome foci (*Liu et al., 2019b*) is compatible with a model where RPG-mediated recruitment of this exocyst subunit could contribute to restrict the tethering of exocytotic vesicles at a single focal site to enable rapid tip growth and IT extension. In the absence of functional RPG, the slow growth caused by unrestricted secretion might negatively feedback on membrane polarization, as shown in yeast and fungi, where reduced growth rates have been linked to the destabilization of polar domains (*Haupt et al., 2018*). Such depolarization might, in turn, alter the cytoskeleton organization with consequent loss of active polar tip growth. Alongside with an alteration of the microtubule-based connectivity between the IT and the nucleus (*Figure 6A–C*), it is likely that actin rearrangements might as well be affected. While the organization of the actin cytoskeleton during infection thread progression will have to be investigated, several genes encoding for actin remodeling proteins were reported to be important for IT growth (*Liang et al., 2021*; *Hossain et al., 2012*; *Yokota et al., 2009*; *Qiu et al., 2015*; *Miyahara et al., 2010*) and actin dynamics were recently shown to be reduced upon inducible depletion of PI(4,5)P$_2$ in *Arabidopsis* root cells (*Doumane et al., 2021*).

Taken together, our data strongly support the hypothesis that IT polarity requires few specific factors like RPG but predominantly relies on a highly conserved mechanism as suggested earlier (*Gage, 2004*; *Yamazaki and Hayashi, 2015*). While the secretion domain in curled root hairs comprises the entire IC (*Fournier et al., 2015*), the loss of highly restricted focal polarization results in uncoordinated expansion of such compartment and failure to maintain IT elongation. Indeed, while thin and elongated ITs are formed in root hairs of WT over the time period of our live-cell imaging analysis (4–8 dpi), the growth of infection structures developed within root hairs of the *rpg-1* mutant appears retarded and their shape is highly variable and irregular (*Figure 4A*, *Figure 5A–B*, *Figure 5—figure supplement 3A*, *Figure 6A–D*). Continuous multiplication of bacteria within these structures then likely leads to their slow and unguided progression across root hairs (*Figure 1B–D*), as previously suggested (*Arrighi et al., 2008*) and further supported by the progressive disintegration of the EMT array and the increase distancing between the nucleus and the IT tip (*Figure 6A–C*), indicative of aborted active growth. Although polarization of the basal membrane domain in the respective trichoblasts and formation of PITs in the outer cortical cells are unaltered (*Figure 4A*, *Arrighi et al., 2008*), effective infection of nodules is impaired (*Figure 1—figure supplement 1B*), likely because a strictly defined time window that allows transcellular IT growth through recently divided cells is missed by these slowly growing ITs in the *rpg* mutant. While the molecular nature of this time window remains unknown, this temporal confinement of cellular infectability coupled with the highly controlled and paced growth of ITs represents an essential layer of host control over intracellularly progressing microbes.

# Materials and methods
## Plant material and phenotypic analysis

*M. truncatula* wild-type ecotypes R108, Jemalong A17, and two *rpg* mutant alleles were used in this study. *rpg-1* is a previously described EMS mutant bearing a premature stop codon in the third exon of the RPG gene (*Arrighi et al., 2008*). Seeds of this allele were kindly provided by Clare Gough

(Institut National de la Recherche Agronomique de Toulouse, Toulouse, France). *rpg-2* is a *Tnt1* insertion line (NF11990) and was obtained from the Noble Research Institute (OK, USA).

Seeds were scarified by soaking them in sulfuric acid ($H_2SO_4$) 96% for 8 min, then rinsed six times with sterile tap water and surface sterilized with a solution of 1.2% sodium hypochlorite (NaClO) and 0.1% sodium dodecyl sulfate (SDS) for 1 min. After rinsing six times with sterile tap water, seeds were transferred to 1% agar/water plates and vernalized at 4°C in dark for 4–5 days. Seeds were then moved to 24 °C in the dark for 10 hr to induce germination.

For phenotypical analysis of IT morphology, germinated seedlings were transferred to pots containing a mixture (1:1 v/v) of quartz sand (grain size 0.1–0.5 mm, Sakret) and vermiculite (0–3 mm, Ökohum), inoculated with *S. meliloti* (Sm2011, $OD_{600}$=0.001, 5 ml per pot) and grown in a controlled environment chamber at 24 °C with a 16/8 hr light/dark photoperiod, and 70% humidity. Pots were watered twice a week with tap water and fertilized once a week with a ¼ Hoagland solution containing 0.1 mM $KNO_3$. Plants were harvested at 35 dpi, the root system was excised and fixed in 4% PFA (paraformaldehyde). Prior to imaging, roots were washed three times with 1 x PBS (phosphate-saline buffer). Infected root hairs were imaged using a confocal microscope and measurement of the infection thread diameter was performed on single stacks from Z-scans using ImageJ/(Fiji) software (*Schindelin et al., 2012*).

For phenotypic analysis of root hair diameter, germinated seedlings were transferred to sterile plates containing solid Fahräeus medium supplemented with 0.5 mM $NH_4NO_3$ and grown vertically in a controlled environment chamber at 24 °C with a 16/8 hr light/dark photoperiod with the root system shaded. After seven days, a 3 mm root segment containing the root tip was excised and mounted in liquid Fahräeus medium. Bright-field images of the root tip and root hairs in the maturation zone were acquired with a Zeiss Apotome microscope using a 5 x objective and a 25 x water immersion objective, respectively. Root hair diameter was measured on single stacks from Z-scans of the maturation zone using ImageJ/(Fiji) software (*Schindelin et al., 2012*).

For morphological examination of cortical infection threads, germinated seedlings were transferred to sterile plates containing solid Fahräeus medium supplemented with 0.1 mM $NH_4NO_3$ and 1 mM of the ethylene-inhibitor aminoethoxyvinylglycine (AVG). Plants were grown vertically in a controlled environment chamber at 24 °C with a 16/8 hr light/dark photoperiod for 1 week before inoculating them with *S. meliloti* (Sm2011, $OD_{600}$=0.01). At 14 dpi the plant root system was excised, fixed in 4% PFA under vacuum for 15 min, and then washed twice with 1 x PBS. Afterward, roots were covered with ClearSee solution (*Ursache et al., 2018*) and incubated for 7–10 days. Prior to imaging, roots were stained with Calcofluor white for 45 min, washed twice with ClearSee, and incubated for 1 hr in the last wash. Cortical infection threads were then imaged using a confocal microscope.

For phenotypic analysis of nodulation capacity, germinated seedlings were transferred to pots containing Zeolite substrate (50% fraction 1.0–2.5 mm, 50% fraction 0.5–1.0 mm, Symbiom), inoculated with *S. meliloti* (Sm2011 pXLGD4, GMI6526, $OD_{600}$=0.02, 10 ml per pot) and grown in a controlled environment chamber at 22 °C with a 16/8 hr light/dark photoperiod, and 70% humidity. Pots were watered once a week with tap water and fertilized once a week with Fahräeus medium (0.5 mM $NH_4NO_3$). Plants were harvested at 21 dpi.

## Hairy root transformation and rhizobial inoculation

Composite *M. truncatula* plants were obtained following the previously described procedure (*Boisson-Dernier et al., 2001*). In brief, transgenic *A. rhizogenes* (ARqua1) carrying the plasmid of interest were grown in liquid LB medium (5 ml) supplemented with appropriate antibiotic for 24 hr until reaching an $OD_{600}$ of approximately 0.5–0.7. 300 µl of this culture were then spread on a plate containing solid LB medium with antibiotics and incubated at 28 °C in dark for 48 hr. The root meristem of germinated seedlings was removed with a scalpel and the wounded part was dragged on the *A. rhizogenes* solid culture before transferring seedlings to plates containing solid Fahräeus medium supplemented with 0.5 mM $NH_4NO_3$. Transformed seedlings were grown vertically in a controlled environment chamber at 22 °C in dark for 3 days and for additional 4 days with a 16/8 hr light/dark photoperiod providing shading to the root system. Afterwards, composite plants were transferred to new solid Fahräeus plates (0.5 mM $NH_4NO_3$) and grown for 10 days at 24 °C with a 16/8 hr light/dark photoperiod. Transformed roots expressing the fluorescent selection marker were selected using a stereo microscope, untransformed roots were excised with a scalpel, and composite plants were transferred to either

solid Fahräeus medium supplemented with 0.1 mM $NH_4NO_3$ (live-cell imaging of root hairs) or to pots (complementation assays).

*Sinorhizobium meliloti* (Sm2011) was grown in liquid TY medium (5 ml) supplemented with appropriate antibiotics for 3 days at 28 °C. 100 µl of this culture was used as inoculum for a fresh culture (5 ml), which was grown for 24 hr at 28 °C. A bacterial pellet was then obtained by centrifugation at 3000 rpm for 10 min, washed once with liquid Fahräeus medium (0.1 mM $NH_4NO_3$), and resuspended in the same medium to reach a final $OD_{600}$=0.01 (inoculation of composite plants in plates, 200 µl per plant) or $OD_{600}$=0.001 (inoculation in pots, 5 ml per plant).

### Construct design

Constructs used in this study were designed and assembled using Golden Gate cloning (*Binder et al., 2014*; *Weber et al., 2011*). All promoters, coding sequences, and genomic sequences used were synthesized by Life Technologies. When internal BsaI or BpiI sites were present, individual point mutations were introduced in silico; in the case of coding and genomic sequences, silent mutations were produced. A list of all L1 and L2 constructs including module composition, fluorophore linkers, and tagging site is provided as a supplementary file (*Supplementary file 1*).

For RPG, a sequence 1777 bp upstream of the start codon was used as a promoter, as described earlier (*Arrighi et al., 2008*). The RPG coding sequence was used for complementation assays, subcellular localization studies, and co-immunoprecipitation assays. RPG fragments corresponding to different deletion derivatives were synthesized by Life Technologies.

For VPY, EXO70H4, and LIN, constructs were designed as described earlier (*Liu et al., 2019b*).

To analyze the subcellular localization pattern of NPL and for fluorescent labeling of microtubules, the coding sequences of NPL and SYMTUB were driven from their 2038 bp and 1555 bp promoters, respectively. Sequences of 2xPH$^{PLC}$ and 2xPH$^{FAPP1}$, and of 2xPH$^{PLC-mut}$ were designed according to *Simon et al., 2014* and (*Yagisawa et al., 1998*), respectively, synthesized by Life Technologies, and expressed under the control of the 1106 bp *ENOD11* promoter.

The sequence data from RPG, VPY, EXO70H4, LIN, NPL, and SYMTUB can be found in phytozome (https://phytozome.jgi.doe.gov/) with the following gene ID: RPG (Medtr1g090807), VPY (Medtr6g027840), EXO70H4 (Medtr4g062330), LIN (Medtr1g090320), NPL (Medtr3g086320), SYMTUB (Medtr8g107250). The sequences of 2xPH$^{PLC}$ and 2xPH$^{FAPP1}$ biosensors can be downloaded at http://www.ens-lyon.fr/RDP/SiCE/PIPline.html.

### Complementation assays

Composite *M. truncatula* plants expressing the constructs of interest were transferred to pots containing a mixture of quartz sand and vermiculite (1:1 v/v), inoculated with *S. meliloti* (Sm2011, $OD_{600}$=0.001, 5 ml per pot) and grown in a controlled environment chamber at 24 °C with a 16/8 hr light/dark photoperiod, and 70% humidity. Pots were watered twice a week with tap water and fertilized once a week with ¼ Hoagland solution containing 0.1 mM $KNO_3$. Plants were harvested at 21 dpi. Transformed roots exhibiting strong expression of the fluorescent marker were selected, excised, and fixed in 4% PFA (paraformaldehyde) under a vacuum for 15 min. After washing the roots twice with 1 x PBS (phosphate-saline buffer), the roots were covered with ClearSee solution (*Ursache et al., 2018*), and incubated for 7–10 days. ClearSee solution was refreshed every two days. Prior to imaging, roots were stained with Calcofluor white for 45 min, washed twice with ClearSee and incubated for 1 hr in the last wash. Infected root hairs were imaged using a confocal microscope and measurement of the infection thread diameter was performed on single stacks from Z-scans with ImageJ/(Fiji) software (*Schindelin et al., 2012*).

### Live-cell imaging of root hairs

Composite plants growing on solid Fahräeus medium supplemented with 0.1 mM $NH_4NO_3$ were inoculated with *S. meliloti* (Sm2011) by adding 200 µl of bacterial suspension ($OD_{600}$=0.01) on each root system. Prior to inoculation, the position of the apex of transformed roots was marked on plates. 4–8 dpi, root segments that had grown below the mark were excised, mounted in water, and root hairs were imaged with a confocal microscope.

### Confocal Laser-Scanning Microscopy (CLSM) and Fluorescence Lifetime Imaging Microscopy (FLIM)

All imaging was performed using a Leica SP8 FALCON FLIM confocal microscope equipped with a 20 x/0.75 and a 40 x/1.1 water immersion lens (Leica Microsystems, Mannheim, Germany). A pulsed

White Light Laser (WLL) was used as an excitation source. GFP and mCitrine were excited at 488 nm and the emission was collected at 500–550 nm. mCherry and mScarlet were excited at 561 nm and emission was detected at 590–640 nm. To excite Calcofluor white, a 405 nm laser diode was used and emission was detected at 425–475 nm.

For FLIM, a Leica GaAsP-hybrid detector was used to collect fluorescent emission. Different channels were acquired in sequential scanning between frames. A pulse repetition rate of 80 mHz was used with the exception that the GFP/NowGFP channel (images in *Figure 3D* and *Figure 3—figure supplement 5*) and the mCherry/mScarlet channel (images in *Figure 3—figure supplement 6A*, *Figure 5B*, and *Figure 5—figure supplement 6*) were acquired using a 40 mHz pulse repetition rate, to allow recording of the full fluorescence decay of NowGFP and mScarlet. Fluorescence emission was acquired from single focal planes using 20 scan repetitions for each channel.

## Intensity-based and FLIM-based image analysis

All intensity-based image processing and analysis were performed using ImageJ/(Fiji) software (*Schindelin et al., 2012*). The Imaris software (Bitplane AG, Switzerland) was specifically used to generate 3D projections and volume renderings. For analysis of the microtubule patterning, z-stacks were deconvolved with Huygens Essential version 22.04 (Scientific Volume Imaging, The Netherlands), using the CMLE algorithm, with Acuity: –60 and 20 iterations.

FLIM-based images were analyzed using LAS X SP8 Control Software (Leica Microsystems GmbH). Global fitting of the intensity decay profile using n-exponential reconvolution was performed to separate major fluorescent components within each channel and calculate their lifetime. The number of components (n) used for curve fitting was determined according to the evaluation of the chi-squared ($\chi 2$) value (*Lakowicz, 2006*) and a threshold of 30 photons was applied to generate the final images, unless otherwise stated. Components with lifetime value <1 ns representing autofluorescent species accumulating in plant cell walls (*Donaldson, 2020*; *Heskes et al., 2012*) were subtracted from each channel.

All data on reconvolution of decay profiles of images presented in the article are provided as source files.

Since global fitting does not always allow to resolve and characterize the decay profile of pixels constituting minor populations within an image (*Ranjit et al., 2018*), the decay profile of fluorescent emission of regions of interest (ROIs) selected on membrane domains was additionally analyzed for the analysis and quantification of PI(4,5)P$_2$ membrane enrichment. n-exponential fitting of the decay curve was carried out to resolve lifetime values associated with each ROI (*Figure 5—figure supplement 1*). Reconvolution data from all images and selected ROIs analyzed are provided as source files. Further, the raw decay profile of each ROI was analyzed using the Phasor approach to obtain a Phasor fingerprint whose position describes the fluorescent emission at that region and reflects its composition with respect to the relative abundance of fluorescent species (*Figure 5—figure supplement 1*) (see *Ranjit et al., 2018* and *Malacrida et al., 2021* for a detailed description of the Phasor approach). Phasor plots were generated using a second harmonic, a threshold of 22 photons, and a median of 15. To visualize and compare the Phasor fingerprint of ROIs selected in multiple images from different genotypes, the center of mass of Phasor images depicting the pixel populations originating from each ROI was calculated images using ImageJ/(Fiji) (*Schindelin et al., 2012*) and the obtained XM and YM coordinates were plotted on a graph (*Figure 5—figure supplement 1*). A Mann-Whitney non-parametric test was performed to detect statistically significant differences between XM and YM values.

For the quantification of PI(4,5)P$_2$ and PI4P membrane enrichment, the mean fluorescence intensity of freehand ROIs drawn on the IT tip and on the plasma membrane of infected root hairs were measured on images of single fluorescent components from each channel, obtained from global fitting of the intensity decay profile (threshold of 20 photons) and subtraction of components with lifetime <1 ns (both channels) and >1.7 ns (channel 2). The mean intensity of a 2.5 × 2.5 µm square ROI outside of the infected root hair cell was measured as a background value, subtracted from each ROI and the mean gray value on the IT tip was divided by the mean gray value on the plasma membrane [$(Int_{IT\ tip} - Int_{bkgd}) / (Int_{PM} - Int_{bkgd})$] for each component to obtain an index of the enrichment of each phosphoinositide at the IT tip relative to the plasma membrane. Statistically significant differences were detected using a Mann-Whitney non-parametric test.

## Statistical analysis

All statistical analysis and generation of graphs were performed using GraphPad Prism software (GraphPad Software Inc). Data were subjected to a normality test to determine the statistical method to apply. Kruskal-Wallis followed by Dunn's post-hoc test or Mann-Whitney was applied as non-parametric tests. Raw data and results of the statistical analysis are provided as source files linked to the corresponding figure.

## Transformation of *N. benthamiana* leaves, protoplast isolation, and coimmunoprecipitation assay

Transgenic *Agrobacterium rhizogenes* (ARqua1) carrying the plasmid of interest were grown in liquid LB media supplemented with appropriate antibiotics for 24 hr at 28 °C. Bacterial cultures were pelleted by centrifugation (4000 rpm), resuspended in Agromix (10 mM MgCl$_2$; 10 mM MES/KOH pH 5.6; 150 uM Acetosyringone), and mixed in selected combinations to reach a final OD$_{600}$ of 0.3 (for GFP and mCherry, 0.1). An *Agrobacterium tumefaciens* (GV3101) carrying a plasmid for expression of the silencing suppressor p19 was added to each mix (final OD$_{600}$=0.1), which was then incubated at room temperature (RT) in the dark for 2 hr. The second and third leaves of 4–5 weeks old *N. benthamiana* plants were infiltrated on the abaxial side using a 1 ml syringe. Since the majority of RPG remained associated with the cell debris fraction when using total protein extracts from leaf tissue, we decided to carry out the extraction after obtaining protoplasts from leaves. 72 hr post-infiltration leaves were harvested and protoplasts were isolated as previously described (*Su et al., 2023b*). Isolated protoplasts were directly lysed in 1 ml protein extraction buffer (50 mM Tris-HCl (pH 7.5), 150 mM NaCl, 10% (v/v) glycerol, 2 mM EDTA, 5 mM DTT, 1 mM phenylmethylsulfonyl fluoride (PMSF), Protease Inhibitor Cocktail (Roche), and 1% (v/v) IGEPAL CA-630 (I3021, Sigma-Aldrich) and incubated for 1 hr at 4 °C. Lysates were centrifuged at 13,000 rpm for 30 min (at 4 °C) to obtain the supernatant fractions, which were then immunoprecipitated using RFP-Traps Magnetica agarose (rta, Chromotek) at 4 °C for 90 min. After incubation with either RFP-Traps or GFP-Traps, samples were washed five times with washing buffer (50 mM Tris-HCl (pH 7.5)), 150 mM NaCl, 10% (v/v) glycerol, 2 mM EDTA, Protease Inhibitor Cocktail (cOmplete, Mini, EDTA-free, 04693159001, Roche), and 0.5% (v/v) IGEPAL CA-630 (I3021, Sigma-Aldrich)). To release the proteins, 40 µl of washing buffer and 10 µl 5 x protein loading buffer were added, and samples were heated for 10 min at 95 °C. 10 µl of each sample were then loaded on a 12% (w/v) SDS gel, run for ~90 mins (100 V, RT), and transferred overnight at 30 V (at 4 °C) to a polyvinylidene difluoride (PVDF) membrane (Immobilon-P, pore size 0.45 µm, IPVH00010, Carl Roth). Transferred membranes were blocked with 5% (w/v) milk for 30 min at RT before being hybridized with the corresponding anti-GFP (632381, Takara) or anti-mCherry (632543, Takara) antibodies at dilutions of 1:3000 and 1:2000, respectively, for 2 hr at RT. Before incubating with the secondary antibody (anti-mouse, A-4415; anti-rabbit, A-6514; Sigma-Aldrich) (1 hr, RT), membranes were washed three times for 10 min with TBST buffer. Membranes were developed following three washes of 10 min each with TBST buffer.

## Live-cell imaging of *N. benthamiana* leaf epidermal cells

4–5 weeks old *N. benthamiana* plants were transiently transformed with transgenic *A. rhizogenes* (ARqua1) carrying the construct of interest. Bacterial cells were resuspended in Agromix to reach a final OD$_{600}$ of 0.5 when the *A. rhizogenes* strain was infiltrated alone, or of 0.3 when co-infiltrated with *Agrobacterium tumefaciens* (GV3101) carrying the plasmid for expression of the silencing suppressor p19 (OD$_{600}$=0.1). 48 hr post-infiltration, two-three leaf disks were excised using a biopsy puncher, mounted in water, and imaged with a confocal microscope.

## Transmission electron microscopy and quantification of nodular IT diameter

Wild-type (A17) and *rpg-1* germinated seedlings were transferred to an aeroponic system (25 liters volume), inoculated with *S. meliloti* (Sm2011, OD$_{600}$=0.01, 5 l), and grown in liquid Fahräeus (0.1 mM NH$_4$NO$_3$) that was replaced weekly. Plants were harvested at 35 dpi. Nodules were cut in half and immediately vacuum infiltrated for 15 min in MTSB buffer containing 4% PFA and 2.5% glutaraldehyde at room temperature. Samples were then left in fixative for 3 hr at room temperature and transferred to 4 °C for overnight. Next, samples were washed with MTSB five times (10 min each) and post-fixed in

a water solution of 1% OsO$_4$ on ice for 2.5 hr. Samples were then washed with water twice for 10 min followed by *in bloc* staining with 1% UrAc in water for 1.5 hr. After washing the samples twice with water for 5 min, they were subjected to dehydration by incubating them for 15 min each in increasing EtOH/water graded series (30%, 50%, 70%, 80%, 95%) and for 30 min twice in 100% EtOH and 100% Acetone. Samples were gradually embedded in Epoxy resin using resin:acetone mixtures (1:3, 1:1, 3:1) 8 hr each, and finally pure resin (three times exchange, 8 hr each). Embedded nodules were polymerized for two days at 60 °C and 70 nm sections were obtained with a Reichert-Jung Ultracut-E microtome. Sections were collected on copper grids and contrasted with 2% uranyl acetate and Reynolds lead citrate solution (*Reynolds, 1963*). Transmission electron microscopy images were obtained using a Philips CM10 (80 kV) microscope coupled to a GATAN Bioscan Camera Model 792 or a Hitachi 7800 TEM operated at 100 kV and coupled to a Xarosa CMOS camera (Emsis).

The diameter of nodular infection threads was measured on TEM images using ImageJ/(Fiji) software (*Schindelin et al., 2012*). A freehand ROI was drawn around the matrix surrounding bacteria contained within cell wall-bounded infection thread structures to define the center of the infection thread area. The infection thread diameter was then measured by drawing a line connecting the borders of the area and crossing its center perpendicularly to the longest axis.

## Materials availability statement

All materials can be requested from the corresponding authors at any time.

## Acknowledgements

We would like to thank Carla Brillada, Marco Trujillo, and Franck A Ditengou as well as Eija Schulze and Rosula Hinnenberg, and Pascal Krohn for their experimental support and technical help, respectively, and the entire Ott lab team for the continuous inputs into the project. We would also like to thank Pierre-Marc Delaux and Jean Keller (LRSV, Université de Toulouse) for critically reading the manuscript and providing a comprehensive phylogeny on RPG, respectively, as well as Leonel Malacrida (Advanced Bioimaging Unit, IP Montevideo, Universidad de la República) for the fruitful discussions on Phasor analysis. We also thank the staff of the Life Imaging Center (LIC) in the Hilde Mangold House (HMH) of the Albert-Ludwigs-University of Freiburg for the help with their confocal microscopy resources, and the excellent support in image recording. The microscopes are operated by the Microscopy and Image Analysis Platform (MIAP) and the Life Imaging Center (LIC), Freiburg. TV belongs to the LRSV, which is part of the TULIP LABEX (ANR-10-LABX-41). The *Medicago truncatula* plants utilized in this research project, which are jointly owned by the Centre National De La Recherche Scientifique, were obtained from Noble Research Institute, LLC, and were created through research funded, in part, by a grant from the National Science Foundation, NSF-0703285. FUNDING Engineering Nitrogen Symbiosis for Africa (ENSA) project is currently supported through a grant to the University of Cambridge by the Bill & Melinda Gates Foundation (OPP1172165) and the UK government's Department for International Development (DFID) (TO) Deutsche Forschungsgemeinschaft (DFG, German Research Foundation) 431626755 (TO) DFG under Germany's Excellence Strategy grant CIBSS – EXC-2189 – Project ID 39093984 (TO) China Scholarship Council (CSC) grants 201708080016 (CS) DFG project number 414136422 (CLSM; TO), DFG project number 426849454 (TEM; TO)

## Additional information

### Funding

| Funder | Grant reference number | Author |
|---|---|---|
| Bill and Melinda Gates Foundation | OPP1172165 | Thomas Ott |
| Deutsche Forschungsgemeinschaft | 431626755 | Thomas Ott |
| Deutsche Forschungsgemeinschaft | 39093984 | Thomas Ott |

| Funder | Grant reference number | Author |
|---|---|---|
| Deutsche Forschungsgemeinschaft | 414136422 | Thomas Ott |
| Deutsche Forschungsgemeinschaft | 426849454 | Thomas Ott |
| China Scholarship Council | 201708080016 | Chao Su |

The funders had no role in study design, data collection and interpretation, or the decision to submit the work for publication.

## Author contributions

Beatrice Lace, Conceptualization, Data curation, Formal analysis, Supervision, Funding acquisition, Investigation, Visualization, Writing - original draft, Project administration, Writing – review and editing; Chao Su, Daniel Invernot Perez, Morgane Batzenschlager, Sabrina Egli, Investigation, Writing – review and editing; Marta Rodriguez-Franco, Tatiana Vernié, Investigation, Methodology, Writing – review and editing; Cheng-Wu Liu, Investigation; Thomas Ott, Conceptualization, Supervision, Funding acquisition, Investigation, Writing - original draft, Project administration, Writing – review and editing

## Author ORCIDs

Beatrice Lace ![orcid] http://orcid.org/0000-0002-4732-573X
Marta Rodriguez-Franco ![orcid] http://orcid.org/0000-0003-1183-2075
Cheng-Wu Liu ![orcid] http://orcid.org/0000-0002-6650-6245
Thomas Ott ![orcid] http://orcid.org/0000-0002-4494-9811

## Decision letter and Author response

Decision letter https://doi.org/10.7554/eLife.80741.sa1
Author response https://doi.org/10.7554/eLife.80741.sa2

# Additional files

## Supplementary files

• Supplementary file 1. List of constructs used in this study.

• MDAR checklist

## Data availability

All data generated or analysed during this study are included in the manuscript and supporting files.

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
