## [Editor Report]

This work addresses a fundamental question in symbiosis, placing a classic nodulation defective mutant (rpg) into a plausible protein complex and establishing a hierarchy of "infectosome" assembly. It will be of particular interest to cell biologists and those studying host-microbe interactions. The study includes compelling microscopy data for subcellular localization of components during the establishment and maintenance of infection and includes new FLIM-based imaging techniques to distinguish signals from closely associated domains in plant cells.

---

## [Decision Letter]

**Decision letter after peer review:**

Thank you for submitting your article "RPG acts as a central determinant for infectosome formation and cellular polarization during intracellular rhizobial infections" for consideration by *eLife*. Your article has been reviewed by 3 peer reviewers, one of whom is a member of our Board of Reviewing Editors, and the evaluation has been overseen by Detlef Weigel as the Senior Editor. The following individual involved in the review of your submission has agreed to reveal their identity: Marie-Cécile Caillaud (Reviewer #3).

Three reviewers agreed that the work was fundamentally strong and interesting, but each also noted areas that could be clarified, and there was some concern about lack of quantification/replication in some experiments. The authors should either provide/present additional evidence, or temper their conclusions and claims.

1) Quantification of several cellular and tissue phenotypes is missing. For example, in Figure 5, please provide some other quantification of the MT and nuclear defects in rpg, or at least a few other representative images as a figure supplement. Please also address the quantification issues brought up by Reviewer 3 major points 1 + 3.

2) Replication of experiments and sample size is not always included. This information should be included in the legends and/or text for each. For replicated experiments, or those for which sample size is too small to make robust statistical conclusions, text and conclusions should be tempered to recognize this limitation.

3) In Co-IP experiments, the signals are rather weak for the interacting partner. The authors should address why the experiments were done in the manner they were (i.e., why did they isolate protoplasts, rather than directly extracting proteins from whole leaves. Surely most samples are lost during protoplasting). Also, co-IP results are typically validated both ways (not only A IPs B, but B also IPs A). Was this done? If not, the authors should temper the conclusions about interactions.

4) Help the reader parse complicated information by provide a diagrammatic model for the different molecular components (lipids, cytoskeleton, protein) they studied in the context of this biotic interaction.

*Reviewer #1 (Recommendations for the authors):*

Overall, I found this an exciting and engaging story. It places a classic nodulation defective mutant into a plausible protein complex and provides solid support for its cellular role in maintaining the infection thread through high-quality imaging and phenotypic analysis. This work established a hierarchy of "infectosome" assembly using genetics and localization of components. I have no major experimental or data analysis concerns. In general, the figures are well laid out and the images are of excellent quality, so the data are easy to evaluate. The sample sizes and statistical measurements are appropriate for the conclusions the authors draw.

*Reviewer #2 (Recommendations for the authors):*

I enjoyed reading this meticulous work. It is very informative and the findings will be highly relevant to the field. However, I also have a number of puzzling observations. I raise them here in the order they appear:

1. In Figure 3A, panels on the right half show the localization inside the infection thread. However, the close-up (in particular panel 2 and 3) does not exactly match the dotted area in panel 1. Several GFP dots, representing bacteria, are missing in the close up. Did the authors happen to use different optical sections for the lower- and high-magnification images?

2. On Line 199 the authors stated that "IT tip-associated puncta encased the growing IT tip". However in Figure 3-supplement 1A the RPG punctum does not look like a case of any sort. This seems an unnecessary overstatement.

3. In Figure 3-supplement 3D, why does the mCherry channel appear to be white even in the merged panel? This makes it hard to assess the claim that RPG-NT is mainly in the nucleus.

4. In the GFP panel of Figure 3B, how did the authors differentiate the signal of VPY-GFP from that of 3xNowGFP? I assume the latter is used as a transformation marker. Does the confocal settings avoid or exclude fluorescence from NowGFP? This is important, as it seems to be the basis for concluding that VPY-GFP "was also diffusely visible in the cytoplasm".

5. The co-IP signals are rather weak for the interacting partner. Most often, co-IP results are validated both ways (not only A IPs B, but B also IPs A). Are the proteins expressed well? If they are, I wonder why the authors isolated protoplasts, rather than directly extracting proteins from whole leaves. Surely most samples are lost during protoplasting.

6. The rpg mutant does form elongating infection threads, albeit highly irregular ones. Yet in Figures 3E, 5A, and 5B, one cannot see the file of bacteria along the mutant root hair. The structure shown here in the rpg mutant is more akin to infection chambers. In 4A, the mutant seems to be just starting an infection thread. Why did the authors not choose elongating infection threads like they did in Figures 1B and 2B?

7. The distinct puncta of VPY disappears in the rpg mutant (Figure 3-supplement 5). Can the authors tell whether the cause is diffuse localization, reduced total protein levels, or reduced transcription in the absence of RPG?

*Reviewer #3 (Recommendations for the authors):*

1. The authors report larger infection threads in rpg mutants than in WT. Could the morphology of uninfected root hair in rpg mutant explain this phenotype? Quantification of the diameter of noninfected RH in WT and rpg mutants should allow the authors to rule out this hypothesis. Moreover, the author reported that TEM micrographs of nodule infected cells revealed "larger IT structures present in nodule tissues of rpg-1 compared to WT but WT-like cell organization and symbiosome morphology": quantification is needed here.

2. To gain more insights into the structural requirements for RPG functionality, the authors generated a series of truncation derivatives and tested them for their capability to genetically complement rpg-1. They showed that the major molecular determinant conferring functionality to RPG resides in its coiled-coil domain. It is not clearly stated how the authors check that the constructs were expressed? It is particularly important for two variants that do not complement the rpg mutant. The authors should explain how they assessed that the constructs were stable and therefore the phenotype observed is not due to an absence of the truncated version but rather to a loss in its activity. It will be appreciated to have the number of independent transformants in the legend of the figure or in the panels (rather than in the material and method).

3. The authors clearly showed that RPG protein has a punctate localization. mCherry-RPG clearly co-localized with VPY-GFP in punctate structures located at the tip of the infection threads and in the cytoplasm connecting the nucleus to the infection threads tip of infected root hairs. However, quantifications are missing for all the data presented in Figure 3A-C, Figure 3—figure supplement 1A-D, Figure 3—figure supplement 2, Figure 3—figure supplement 3A-D, Figure 3—figure supplement 4A-D.

4. The author showed that RPG co-purified with EXO70H4 and VPY indicating that these proteins indeed belong to the same complex. Moreover, they showed that the major molecular determinant conferring functionality to RPG resides in its coiled-coil domain. Does the CC domain important for RPG interaction with VAPYRIN (VPY)/mCherry-EXO70H4? Furthermore, there is no clear indication of the number of times this experiment has been replicated.

5. While no quantification is presented, the authors showed with clear illustrations that the mCherry-RPG-NT construct which does not complement the rpg mutant remained diffuse in the cytoplasm and preferentially accumulated in the nucleus of transformed root hairs. It would be interesting to repeat this analysis (mCherry-RPG-CC, mCherry-RPG-tCC, mCherry-RPG-NT) in WT plants: These complementary analyses will allow the authors to further evaluate if the function of RPG depends on its localization or on its interaction with protein partners.

6. In Figure 4, they reported that in uninfected growing root hairs, PI(4,5)P2 was enriched at the apical membrane domain while at the onset of infection PI(4,5)P2 was locally and specifically enriched at the membrane surrounding invading bacteria emerging from the infection chamber. The observations presented in Figure 4B and Figure 4—figure supplement 5 should be quantified. Did the authors try to over-express RPG and check for a potential increase in membrane polarity? How do these increases in PIP2 polarity occur? Is it via the transcriptional regulation of the PIP2 metabolism or via the trafficking/activity of the PIP2 regulating enzymes? Some pieces of information should be added at least in the discussion of the paper.

7. They further observed that local PI(4,5)P2 enrichment also occurred on the basal membrane of root hairs hosting polarized infection threads and on the apical membrane of underlying cortical neighboring cells. It is not clear to me how the authors know that this membrane belongs to the infected cell and is not part of another cell that is about to initiate a root hair. Imaging in 3D will help to understand the topology here.

8. Since nuclear movement appeared uncoupled from the infection threads' growth in rpg, the authors tested the implication of the MTs in this phenomenon. Even if the authors are citing other studies showing the role of MTs, it is not clear to me why the actin cytoskeleton was excluded from this analysis. Maybe the authors should consider discussing it, in particular since the actin cytoskeleton is important for polar growth and might be linked to PIP2 polarity.

9. By comparing microtubule organization in WT and rpg root hairs harboring the infection threads using live-cell imaging, they reported the loss of nuclear-mediated IT guidance occurs in rpg "in the majority of the case", "often". Please quantify more deeply the results obtained in Figure 5A (in particular in the corresponding text of the result section). Why the nucleus is not visible in magenta in the right panels?

10. Regarding the polar secretion of NPL which requires functional RPG, is there a direct role of the membrane polarization (or PIP2) in the exocytosis of cell wall modifying enzyme NODULE PECTATE LYASE (NPL)? Maybe some information could be added in the discussion of the manuscript about the general knowledge from the literature.

11. The authors should consider adding a model for the different molecular components they studied in the context of this biotic interaction.

---

## [Author Response]

1) Quantification of several cellular and tissue phenotypes is missing. For example, in Figure 5, please provide some other quantification of the MT and nuclear defects in rpg, or at least a few other representative images as a figure supplement. Please also address the quantification issues brought up by Reviewer 3 major points 1 + 3.

We now provide quantifications for all experiments and apologize for not having done so in full in the previous version and that we used different ways (in figure numbers, legends, text) of stating the quantifications, which certainly gave the impression that some of the experiments hadn’t been quantified appropriately. For this, we have repeated several experiments to increase sample size and re-evaluated all our data to ensure maximum reliability. For details, please see the individual responses below.

2) Replication of experiments and sample size is not always included. This information should be included in the legends and/or text for each. For replicated experiments, or those for which sample size is too small to make robust statistical conclusions, text and conclusions should be tempered to recognize this limitation.

Also for sample size and number, we are now much more precise and again apologize for not having been fully consistent in the first version. Corresponding statistical analyses have now been included where missing.

3) In Co-IP experiments, the signals are rather weak for the interacting partner. The authors should address why the experiments were done in the manner they were (i.e., why did they isolate protoplasts, rather than directly extracting proteins from whole leaves. Surely most samples are lost during protoplasting). Also, co-IP results are typically validated both ways (not only A IPs B, but B also IPs A). Was this done? If not, the authors should temper the conclusions about interactions.

We have now repeated the co-IP analysis and also used the reciprocal approach (Figure 3—figure supplement 7). We also provide explanations for the comparable weak bands and on our experimental approach. Please see the corresponding reviewer comment for more details.

4) Help the reader parse complicated information by provide a diagrammatic model for the different molecular components (lipids, cytoskeleton, protein) they studied in the context of this biotic interaction.

We have now added graphical models for most experiments to guide the reader through these complex data. This indeed increased understandability of the figures significantly.

Reviewer #2 (Recommendations for the authors):I enjoyed reading this meticulous work. It is very informative and the findings will be highly relevant to the field. However, I also have a number of puzzling observations. I raise them here in the order they appear:1. In Figure 3A, panels on the right half show the localization inside the infection thread. However, the close-up (in particular panel 2 and 3) does not exactly match the dotted area in panel 1. Several GFP dots, representing bacteria, are missing in the close up. Did the authors happen to use different optical sections for the lower- and high-magnification images?

The two images are maximum projections of two different z-stack series, with the close-up being acquired subsequently, and at higher magnification. Therefore, the optical sections are not perfectly matching. We have now replaced this image with an image where the nucleo-cytoplasmic signal of RPG is detected (Figure 3A-IT and Figure 3—figure supplement 1C-D), as more representative of RPG patterning. As before, the close-up was acquired subsequently at higher magnification, and we have now included this information in the figure legend.

2. On Line 199 the authors stated that "IT tip-associated puncta encased the growing IT tip". However in Figure 3-supplement 1A the RPG punctum does not look like a case of any sort. This seems an unnecessary overstatement.

We rephrased our statement referring to IT tip-associated puncta in Figure 3—figure supplement 1A as “a patch juxtaposed to the growing IT tip ahead of progressing bacteria”.

3. In Figure 3-supplement 3D, why does the mCherry channel appear to be white even in the merged panel? This makes it hard to assess the claim that RPG-NT is mainly in the nucleus.

The apparent white coloring of the mCherry channel in the merge panel is due to the fact that mCherry-RPG-NT (green) perfectly co-localizes with the nuclear-localized tandem CPF (magenta) in the nucleus, giving white as a sum of the two opposite colors in the merge image. To further support the claim, and facilitate a comparison between the localization pattern of RPG-FL and RPG-NT, we have implemented in Figure 3—figure supplement 4 a plot of the intensity profile measured across the nucleus of infected root hairs presented in Figure 3—figure supplement 3A and 3D.

4. In the GFP panel of Figure 3B, how did the authors differentiate the signal of VPY-GFP from that of 3xNowGFP? I assume the latter is used as a transformation marker. Does the confocal settings avoid or exclude fluorescence from NowGFP? This is important, as it seems to be the basis for concluding that VPY-GFP "was also diffusely visible in the cytoplasm".

First of all, we apologize as we noticed we did a mistake in annotating the construct in the upper line of the panel, where pAtUbi::2xNowGFP should instead be pAtUbi::NLS-2xNowGFP, since we used a nuclear-targeted tandem NowGFP as transformation marker and not a cytosolic 2xNowGFP. We have now corrected this mistake. Beside this, the separation of the GFP and the NowGFP signal is based on their different lifetimes (GFP ~ 2.5-2.7 ns; Sarkisyan et al., 2015 NowGFP ~ 4.0 ns, Abraham et al., 2015), and we have added this information in the text. Moreover, we have implemented a detailed explanation of how the FLIM-based separation of GFP and NowGFP was performed in a separate supplementary figure (Figure 3—figure supplement 5), supporting the statement that VPY-GFP is diffusely present in the cytoplasm.

5. The co-IP signals are rather weak for the interacting partner. Most often, co-IP results are validated both ways (not only A IPs B, but B also IPs A). Are the proteins expressed well? If they are, I wonder why the authors isolated protoplasts, rather than directly extracting proteins from whole leaves. Surely most samples are lost during protoplasting.

We fully agree that the bands are comparably weak. We have now validated these results by reciprocal pull-downs as suggested (Figure 3—figure supplement 7). While we got reproducible interactions between RPG and EXO70H4, we can pull VPY only when using RPG as a bait. Given the presence of a cytosolic fraction of VPY we currently believe that only a subfraction of the VPY pool interacts with RPG. It should be noted that all three proteins indeed seemed to be lowly expressed in *N. benthamiana* leaves, as we were unable to detect a signal in a classical input fraction. However, upon immuno-precipitation, we got reliable signals of all three proteins when used as baits. One thing that really hampered any further biochemical analysis is the fact that the RPG protein is extremely difficult to isolate and to handle. The same experience was made by Fang Xie’s lab, who just reported on this in their recent publication (Li et al., 2023; PLoS Genetics). We experienced such difficulties as well when trying to isolate RPG from total protein extracts of leaves. Here, the majority of the protein precipitated with the debris fraction. This is the reason why we decided to first isolate protoplasts, in order to remove the cell wall, and to do the Co-IPs on this type of material.

6. The rpg mutant does form elongating infection threads, albeit highly irregular ones. Yet in Figures 3E, 5A, and 5B, one cannot see the file of bacteria along the mutant root hair. The structure shown here in the rpg mutant is more akin to infection chambers. In 4A, the mutant seems to be just starting an infection thread. Why did the authors not choose elongating infection threads like they did in Figures 1B and 2B?

During our live-cell imaging approach we specifically aimed to image infected root hairs within a similar and early time window (4-8 dpi) in WT and *rpg-1* transgenic roots expressing the different markers. At this early time point, the infection structures formed in *rpg-1* roots are highly irregular and, indeed, more akin to enlarged infection chambers, compared to thin and elongated ITs formed in WT roots, which well correlates with a defect in polarity maintenance. ITs shown in Figure 1B and 2B have been imaged at 35 and 21 dpi, respectively, and their elongated shape is likely the result of passive and unrestricted bacterial multiplication coupled to unpolarized growth occurring over time. To clarify this point, we have added few sentences in the discussion.

7. The distinct puncta of VPY disappears in the rpg mutant (Figure 3-supplement 5). Can the authors tell whether the cause is diffuse localization, reduced total protein levels, or reduced transcription in the absence of RPG?

This is a very interesting question but difficult to address conclusively at the moment. The fact that we can consistently observe a clear and diffuse cytoplasmic signal of VPY in both WT and *rpg-1* transgenic roots expressing *VPY* under its native promoter (Figure 4A and Figure 4—figure supplement 1), suggests that, if present, reduction of VPY either at the transcript or protein level is subtle. Although detection of global VPY transcript or protein abundance from WT and *rpg-1* transgenic roots could be performed, such analyses would not maintain cellular resolution. On the other hand, quantitative measurements of the cytoplasmic signal from VPY in root hairs would not be providing a decisive answer either, and they are likely to be subjected to high variation. To capture small and dynamic variations while maintaining cellular resolution, single-cell based transcriptomic or proteomic approaches would be required, but as such approaches are highly time and resource demanding, we decided to not undertake them.

Reviewer #3 (Recommendations for the authors):1. The authors report larger infection threads in rpg mutants than in WT. Could the morphology of uninfected root hair in rpg mutant explain this phenotype? Quantification of the diameter of noninfected RH in WT and rpg mutants should allow the authors to rule out this hypothesis. Moreover, the author reported that TEM micrographs of nodule infected cells revealed "larger IT structures present in nodule tissues of rpg-1 compared to WT but WT-like cell organization and symbiosome morphology": quantification is needed here.

We have now quantified the diameter of differentiated root hair cells in the maturation zone of *rpg* mutant alleles and their corresponding wild-types in un-inoculated conditions (Figure 1—figure supplement 1A-C), confirming that the IT phenotype is not due to an altered root hair morphology. We also provide quantification of the diameter of nodular ITs developed in WT and *rpg-1* nodules (Figure 1F), supporting our initial statement.

2. To gain more insights into the structural requirements for RPG functionality, the authors generated a series of truncation derivatives and tested them for their capability to genetically complement rpg-1. They showed that the major molecular determinant conferring functionality to RPG resides in its coiled-coil domain. It is not clearly stated how the authors check that the constructs were expressed? It is particularly important for two variants that do not complement the rpg mutant. The authors should explain how they assessed that the constructs were stable and therefore the phenotype observed is not due to an absence of the truncated version but rather to a loss in its activity. It will be appreciated to have the number of independent transformants in the legend of the figure or in the panels (rather than in the material and method).

When conducting these experiments, we consistently observed signals from RPG-FL and all the deletion derivatives in root hairs of transgenic *rpg-1* roots, coinciding with the presence of either thin and elongated ITs (RPG-FL and RPG-CC, Figure 3A-B) or enlarged IT structures (RPG-tCC and RPG-NT, Figure 3C-D), in good correlation with our complementation data (Figure 2A-C). This suggests that the deletion derivatives are sufficiently expressed, although we cannot completely rule out that domain deletion has influenced their expression and/or the protein stability. We have mentioned this in the discussion session.

Each composite plant equals to an independent transformant, as its root system derives from an independent transformation event. We apologize for not having explained this, as also pointed out by reviewer 1. We have now explained the term “composite plant” in the text when mentioned for the first time.

3. The authors clearly showed that RPG protein has a punctate localization. mCherry-RPG clearly co-localized with VPY-GFP in punctate structures located at the tip of the infection threads and in the cytoplasm connecting the nucleus to the infection threads tip of infected root hairs. However, quantifications are missing for all the data presented in Figure 3A-C, Figure 3—figure supplement 1A-D, Figure 3—figure supplement 2, Figure 3—figure supplement 3A-D, Figure 3—figure supplement 4A-D.

For each figure, we now report the number of cells analyzed for each infection stage, together with the number of independently transformed root systems. Frequencies of observation are additionally reported when directly comparing the sub-cellular patterning of deletion derivatives (RPG-FL and RPG-CC). Co-localization data are presented as percentage of discrete signals from RPG and/or VPY/LIN in punctate structures in a separate figure (Figure 3F and Figure 3—figure supplement 6C).

4. The author showed that RPG co-purified with EXO70H4 and VPY indicating that these proteins indeed belong to the same complex. Moreover, they showed that the major molecular determinant conferring functionality to RPG resides in its coiled-coil domain. Does the CC domain important for RPG interaction with VAPYRIN (VPY)/mCherry-EXO70H4? Furthermore, there is no clear indication of the number of times this experiment has been replicated.

We agree that this is an important question but we decided to not extend our co-immunoprecipitation analysis to the RPG deletion derivatives, since we are not able at this moment to fully address biochemically the interaction between RPG and VPY. We recognize this limitation in the result part of the text.

We apologize for not having stated the number of replica for the Co-IP, we have now implemented this information in the legend of the corresponding figure (Figure 3—figure supplement 7).

5. While no quantification is presented, the authors showed with clear illustrations that the mCherry-RPG-NT construct which does not complement the rpg mutant remained diffuse in the cytoplasm and preferentially accumulated in the nucleus of transformed root hairs. It would be interesting to repeat this analysis (mCherry-RPG-CC, mCherry-RPG-tCC, mCherry-RPG-NT) in WT plants: These complementary analyses will allow the authors to further evaluate if the function of RPG depends on its localization or on its interaction with protein partners.

As for all the other data concerning sub-cellular distribution patterns, we now report the number of cells analyzed for each infection stage, together with the number of independently transformed root systems for mCherry-RPG-NT.

We agree that such complementary analysis would integrate our functional studies and we had done explorative experiments assessing the localization of mCherry-RPG-CC and mCherry-RPG-NT in WT transgenic roots. Although preliminary, these observations did not reveal any obvious differential pattern of the two truncated versions in WT vs *rpg-1*. Therefore, we decided to prioritize the analysis of the localization of the deletion derivatives in the *rpg* mutant as more informative when coupled to our functional analysis.

6. In Figure 4, they reported that in uninfected growing root hairs, PI(4,5)P2 was enriched at the apical membrane domain while at the onset of infection PI(4,5)P2 was locally and specifically enriched at the membrane surrounding invading bacteria emerging from the infection chamber. The observations presented in Figure 4B and Figure 4—figure supplement 5 should be quantified. Did the authors try to over-express RPG and check for a potential increase in membrane polarity? How do these increases in PIP2 polarity occur? Is it via the transcriptional regulation of the PIP2 metabolism or via the trafficking/activity of the PIP2 regulating enzymes? Some pieces of information should be added at least in the discussion of the paper.

We have now quantified the differential enrichment of PI4P and PI(4,5)P_2_ in WT vs *rpg-1* (previously Figure 4B and Figure 4—figure supplement 5) by assessing the IT tip versus plasma membrane fluorescence ratio of each biosensor in infected root hairs (Figure 5B), statistically supporting our observations.

We have added a paragraph in the discussion where we discuss how the membrane enrichment of PI(4,5)P_2_ during infection could potentially occur and how RPG could impinge on it. We did not monitor membrane polarity upon ectopic expression of RPG. Although it could be interesting to perform such experiment, we think that, on its own, it would not provide decisive information to unravel the mechanism by which RPG affects membrane polarization. Further experiments with the required depth and robustness would, however, by far exceed the scope of this manuscript.

7. They further observed that local PI(4,5)P2 enrichment also occurred on the basal membrane of root hairs hosting polarized infection threads and on the apical membrane of underlying cortical neighboring cells. It is not clear to me how the authors know that this membrane belongs to the infected cell and is not part of another cell that is about to initiate a root hair. Imaging in 3D will help to understand the topology here.

As requested by the reviewer, we have now investigated this experimentally and added a 3D projection and a 3D volume rendering of the samples shown in Figure 5A-CC in Figure 5—figure supplement 2C and 3B.

8. Since nuclear movement appeared uncoupled from the infection threads' growth in rpg, the authors tested the implication of the MTs in this phenomenon. Even if the authors are citing other studies showing the role of MTs, it is not clear to me why the actin cytoskeleton was excluded from this analysis. Maybe the authors should consider discussing it, in particular since the actin cytoskeleton is important for polar growth and might be linked to PIP2 polarity.

We certainly agree that the actin cytoskeleton is likely playing an important role during IT polarity. We indeed performed preliminary experiments to monitor the actin cytoskeleton in WT vs *rpg-1* infected root. On the one hand, we used Phalloidin on fixed roots to stain actin filaments within root hairs but we could not obtain conclusive and robust results, mainly due to the low amount of infected root hairs maintaining a well-preserved cytological organization after fixation. On the other hand, we tried to visualize actin fibers in live samples by overexpressing the actin binding probe LifeAct, but ectopic expression of this marker seemed to negatively affect infection, and we could not reliably image root hair hosting infection threads neither in WT nor in *rpg-1* transgenic roots.

We have recognized the possible involvement of the actin cytoskeleton and the necessity to investigate it in the discussion.

9. By comparing microtubule organization in WT and rpg root hairs harboring the infection threads using live-cell imaging, they reported the loss of nuclear-mediated IT guidance occurs in rpg "in the majority of the case", "often". Please quantify more deeply the results obtained in Figure 5A (in particular in the corresponding text of the result section). Why the nucleus is not visible in magenta in the right panels?

As requested, we now provide five additional representative pictures of the microtubule patterning in infected root hairs of WT and *rpg-1*, which are presented as a supplementary figure (Figure 6—figure supplement 1). We also provide a more precise quantification of our observations in the corresponding text. In addition, we provide quantification of the distance between the nucleus and the tip of the infection thread, further supporting the loss of nuclear-mediated IT guidance in infected root hairs of the *rpg-1* mutant.

We have changed the images in the right panel (Figure 6A *rpg-1*) with a more representative one, where the nucleus and IT tip are at a distance falling close to the median calculated by our quantification. The previous image has been moved to the supplementary (Figure 6—figure supplement 1). To augment the visibility of the nucleus, we have increased the contrast of the mCherry channel.

10. Regarding the polar secretion of NPL which requires functional RPG, is there a direct role of the membrane polarization (or PIP2) in the exocytosis of cell wall modifying enzyme NODULE PECTATE LYASE (NPL)? Maybe some information could be added in the discussion of the manuscript about the general knowledge from the literature.

While the NPL homolog of soybean has been shown to co-localize with VAMP721d-positive compartments in infected nodule cells (Gavrin et al., 2016, New Phytol.), we are not aware of a direct functional link reported between NPL and membrane polarization during IT growth and, at this stage, we cannot fully resolve the primary reason for the secretion defect. However, and in line with a loss of a PIP2 domain at the IT tip, we strongly believe that targeted vesicle delivery to this site is abolished in the mutant. This hypothesis is strengthened by the fact that we do observe secretion of NPL to the symbiotic apoplastic compartment but this remains unconfined (Figure 6 D-F). Since RPG interacts with EXO70H4 residing within infectosome foci, it is possible that RPG-mediated recruitment of this exocyst subunit could enable the confinement of secretion at a unique focal point. We have now added this to the discussion.

11. The authors should consider adding a model for the different molecular components they studied in the context of this biotic interaction.

We have added graphical summaries of the conclusions drawn from each set of experiments in the respective Figure or panel.